# A 27-country test of communicating the scientific consensus on climate change

Communicating the scientific consensus that human-caused climate change is real increases climate change beliefs, worry and support for public action in the United States. In this preregistered experiment, we tested two scientific consensus messages, a classic message on the reality of human-caused climate change and an updated message additionally emphasizing scientific agreement that climate change is a crisis. Across online convenience samples from 27 countries ($n = 10{,}527$), the classic message substantially reduces misperceptions ($d = 0.47$, 95% CI (0.41, 0.52)) and slightly increases climate change beliefs (from $d = 0.06$, 95% CI (0.01, 0.11) to $d = 0.10$, 95% CI (0.04, 0.15)) and worry ($d = 0.05$, 95% CI (−0.01, 0.10)) but not support for public action directly. The updated message is equally effective but provides no added value. Both messages are more effective for audiences with lower message familiarity and higher misperceptions, including those with lower trust in climate scientists and right-leaning ideologies. Overall, scientific consensus messaging is an effective, non-polarizing tool for changing misperceptions, beliefs and worry across different audiences.

There is near-universal consensus (97–99.9%) in the peer-reviewed scientific literature that the climate is changing as a result of human activity[1–3]. However, the public often considerably underestimates this consensus[4], which is evident even in recent data from six European countries (estimates range from 65% in the United Kingdom to 71% in Ireland)[5]. These misperceptions have several negative consequences that can impede efforts to mitigate climate change[6,7]. People who underestimate the scientific consensus on climate change are less likely to believe in (human-caused) climate change, to worry about climate change and to support public action on climate change compared to those who perceive the scientific consensus more accurately[6–10].

On a more optimistic note, these misperceptions offer an opportunity for low-cost interventions that can be applied at scale. In recent years, communicating the message that 97% of climate scientists agree that human-caused climate change is happening has been one of the most studied strategies to correct misperceptions of the scientific consensus and influence climate change attitudes[11,12]. A large body of experimental studies supports the notion that communicating this scientific consensus can shift consensus perceptions, beliefs in the reality of climate change and human activity as its primary cause,

climate change worry and support for public action[13–25]. In addition, two meta-analyses show that informing people about the scientific consensus can substantially reduce consensus misperceptions (Hedge's $g = 0.56$)[26] and—to a smaller extent—increase several proclimate attitudes (Hedge's $g = 0.09$–0.12)[26,27], namely personal beliefs in and worry about climate change. One of these meta-analyses shows that messaging interventions—including but not limited to scientific consensus messages—had no effect on support for public action[27].

However, the current evidence base relies overwhelmingly on data from Western, democratic, high-income countries. This includes most studies from the United States and other native English-speaking countries (United States, $k = 18$; Australia, $k = 2$; New Zealand, $k = 1$; $k$ is the number of studies in the most recent meta-analysis[26]) as well as two exceptions, one from Japan[15] and one from Germany[28]. Given that climate change requires global action and cooperation[29], the lack of knowledge on whether and to what extent scientific consensus messages can reduce consensus misperceptions and shift climate change attitudes outside the United States and the few studied Western nations is a pivotal gap that needs addressing[12,30–33]. The present study aims to extend the evidence on scientific consensus messaging to include 27 countries on six continents (Asia, Africa, Australia, Europe and

✉ e-mail: sandra.geiger@univie.ac.at; dar56@cam.ac.uk

North and South America). As the public in many countries underestimates the scientific consensus on the reality of human-caused climate change[5,34], a message that emphasizes this consensus is expected to increase consensus perceptions and proclimate attitudes in a diverse, multicountry sample.

Beyond the consensus on the reality of human-caused climate change[5,24,25,28,33,35], climate experts emphasize very high certainty of the adverse consequences of climate change and the urgency of climate action to curb these impacts in the Sixth IPCC report[35]. In line with this, 88% of surveyed IPCC authors report that they think climate change constitutes a crisis[36]. To align communication about the scientific consensus with these more up-to-date climate science assessments and potentially improve its effectiveness, we test a combined message communicating the 97% consensus that human-caused climate change is happening in addition to the 88% agreement that climate change is an urgent matter (a crisis). Such an updated message that emphasizes the negative impacts of climate change and implies the need for public action might prove more effective at increasing belief in climate change as a crisis, climate change worry and support for public action than the classic message[28]. This might be especially useful in contexts where the public consensus on the reality of climate change is high but a substantial proportion still doubts the urgency of climate action[28]. Initial support for the effectiveness of such an updated message comes from the finding that communicating the social consensus on the urgency of climate action increases support for public action on climate change more than a social consensus message on the reality of climate change[37].

While we expect both scientific consensus messages to be overall effective, their effectiveness may depend on several individual-level (message familiarity, trust in climate scientists and political ideology) and country-level (individualism–collectivism and power distance[38]) characteristics. Regarding individual-level characteristics, Bayesian approaches to information processing postulate that people are generally motivated to be accurate and update their beliefs according to relevant information[39], especially when the new information challenges previous beliefs[40,41]. Previous research shows that scientific consensus messages on climate change are more effective for individuals with lower initial consensus perceptions[24,26]. In line with this reasoning, scientific consensus messages might be less effective when individuals are more familiar with the message[42]. Moreover, according to some motivated cognition accounts[43,44], people whose previous worldviews and/or identities are not aligned with a given issue or its implications might not be receptive to scientific (consensus) information on the topic or even revise away from the scientific evidence. In line with this view, trust in climate scientists and political ideology might moderate the effectiveness of any consensus intervention, with those lower in trust and more politically on the right[6] being less likely to accept the scientific consensus on climate change. To date, however, the evidence for such effects is mixed. Although there is some suggestive evidence that those with higher trust in science/scientists might be more receptive to scientific consensus messages[15,45,46], strong evidence using larger samples is lacking to support these patterns. With respect to political ideology, studies show smaller[13], similar[19,22], larger[23,24,31] and even backfiring[47] effects of scientific consensus interventions among conservatives compared to other political groups and the most recent meta-analyses found no moderating effect of political ideology[26,48].

If the effectiveness of the intervention varies across countries, this could be due to country-level differences in individualism–collectivism and power distance[38]. Consensus information is a form of a descriptive norm, and norms have been shown to be more predictive of support for environmental policy in collectivistic cultures compared to individualistic ones[49]. Further considering that scientific consensus messages are an expert norm, it is possible that they are more effective in cultures with higher power distance, where greater weight is given to source expertise/authorities[50–52]. However, more precise predictions are difficult because of the lack of studies on scientific consensus interventions outside the United States[12,30–33].

In sum, this study expands the evidence base on the effectiveness of scientific consensus messaging on climate change in several ways (Table 1). We first test the effectiveness of the classic (reality of human-caused climate change) and the updated (reality of human-caused climate change and crisis) consensus messages across 27 countries. To do so, we focus on the main effects of the interventions on perceptions of the reality consensus and crisis agreement, personal climate change beliefs (reality, human causation and crisis), climate change worry and support for public action in a between-participants design ($H_{1a–e}$ and $H_{2a–g}$). Second, we test whether supplementing the scientific consensus on the reality of climate change with the broad scientific agreement that climate change constitutes a crisis can further increase personal belief in climate change as a crisis, climate change worry and support for public action ($H_{3a–c}$). Third, this study provides an opportunity for a comprehensive, high-powered investigation into individual-level characteristics, such as message familiarity, trust in climate scientists and political ideology, which might moderate the effectiveness of both interventions on reality consensus and crisis agreement perceptions ($H_{4a–c}$ and Q5). Last, we explore whether and to what extent the effectiveness of both messages varies across countries (Q6) and whether country-level characteristics, such as individualism–collectivism and power distance, can predict potential variation (not preregistered). Tests corresponding to each hypothesis are summarized in Supplementary Table 1.

Testing climate change consensus messaging across various countries has ramifications at two levels. Theoretically, it addresses generalizability concerns about the effects of expert norm communication. In addition, well-powered moderation analyses enable us to address conflicting theoretical standpoints (Bayesian information processing and motivated cognition) about human cognition in the face of contested scientific evidence. On a translational level, a messaging approach that is effective across diverse contexts and audiences would provide a general guideline for climate change communication and could thus facilitate a more rapid move toward urgently needed climate policies. If the effectiveness varies according to individual and country-level characteristics, this could inform targeting specific audiences within countries and/or calibrating consensus messaging interventions to different country contexts. However, if consensus messaging is ineffective when tested across a diverse set of countries, this would signal the limits of this intervention and the need to focus on different strategies to mobilize support for climate action.

## Results

### Participants

The analytical sample consisted of 10,527 total participants. Country sample sizes ranged from 9 (Lebanon) to 634 (Germany). In terms of gender, women were slightly more represented than men (female, 57%; non-binary or prefer not to say, 1%). Most of the sample held a university degree (68.1%) and lived in urban areas (81.6%). About one-third of the sample were studying at the time of data collection (33.7%). Demographic overviews per country are presented in Table 2, while population descriptions for each country are available in Supplementary Table 5.

Our recruitment approach is described in detail in the Methods section 'Participant recruitment' and further details about the sample are reported in the section 'Sample details'.

### Overview

The preregistered data analyses focus on three broad questions. First, we test whether the classic and the updated scientific consensus messages can reduce misperceptions and increase climate change beliefs,

**Table 1 | Overview of preregistered research questions and hypotheses**

| Research question | Hypothesis |
|---|---|
| Q1. Is the classic scientific consensus message effective compared to a control message? | $H1_{a-e}$ (main effects: control versus classic consensus). |
| | Compared to the control condition, participants in the classic consensus condition: |
| | (a) perceive a higher scientific consensus that human-caused climate change is happening (controlling for pre-intervention perceptions of the reality consensus), |
| | (b) believe more in the reality of climate change, |
| | (c) believe more in the human causation of climate change, |
| | (d) worry more about climate change and |
| | (e) support public action on climate change more. |
| Q2. Is the updated scientific consensus message effective compared to a control message? | $H2_{a-g}$ (main effects: control versus updated consensus). |
| | Compared to the control condition, participants in the updated consensus condition: |
| | (a) perceive a higher scientific consensus that human-caused climate change is happening (controlling for pre-intervention perceptions of the reality consensus), |
| | (b) perceive higher scientific agreement that climate change is a crisis (controlling for pre-intervention perceptions of the crisis agreement), |
| | (c) believe more in the reality of climate change, |
| | (d) believe more in the human causation of climate change, |
| | (e) believe more that climate change constitutes a crisis, |
| | (f) worry more about climate change and |
| | (g) support public action on climate change more. |
| Q3. Is the updated scientific consensus message more effective than the classic consensus message? | $H3_{a-c}$ (main effects: classic consensus versus updated consensus). |
| | Compared to the classic consensus condition, participants in the updated consensus condition: |
| | (a) believe more that climate change constitutes a crisis, |
| | (b) worry more about climate change and |
| | (c) support public action on climate change more. |
| Q4. Does the effectiveness of the classic consensus message vary by subgroup? | $H4_{a-c}$ (interaction effects: control versus classic consensus). |
| | Controlling for pre-intervention perceptions of the reality consensus, the effect of the classic consensus versus control condition on reality consensus perceptions is moderated by: |
| | (a) message familiarity, such that the message is more effective for those who report lower familiarity with the classic consensus statement and |
| | (b) trust in climate scientists, such that the message is more effective for those who report greater trust in climate scientists. |
| | Controlling for pre-intervention perceptions of the reality consensus, the effect of the classic consensus versus control condition on reality consensus perceptions is not moderated by: |
| | (c) political ideology. |
| Q5. Does the effectiveness of the updated consensus message vary by subgroup? | We planned the following exploratory analyses: |
| | 1. Is the effect of the updated versus control condition on reality consensus perceptions moderated by: |
| | 2. (a) message familiarity (of the classic consensus message), |
| | 3. (b) trust in climate scientists and |
| | 4. (c) political ideology, controlling for pre-intervention perceptions of the reality consensus? |
| | 5. Is the effect of the updated versus control condition on crisis agreement perceptions moderated by: |
| | 6. (a) message familiarity (of the classic and the updated consensus message), |
| | 7. (b) trust in climate scientists and |
| | 8. (c) political ideology, controlling for pre-intervention perceptions of the crisis agreement? |
| Q6. Does the effectiveness of both interventions vary by country? | We planned to explore if the effectiveness of the interventions varies by country. |

worry and support for public action (Q1 and Q2 in Table 1). Second, we investigate whether the updated message is more effective than the classic one at shifting personal belief in climate change as a crisis, climate change worry and support for public action (Q3). Third, we test whether the effectiveness of both interventions varies by several individual and country-level characteristics as well as by country (Q4–Q6 and further exploratory analyses).

To do so, we rely on Bayesian model-averaging approaches[53,54]. Our analyses (Supplementary Information Section 1; Open Science Framework (OSF) https://osf.io/z6quh/) take into account the uncertainty regarding the model structure (for example, constant versus heterogeneous intervention effects across countries). Furthermore, we used Bayesian mixed-effects linear and ordinal regressions, with participants (level 1) nested in countries (level 2), controlling for relevant

**Table 2 | Demographic characteristics across countries**

| Country | N | Mean age (s.d.) | Gender | | | Urbanicity | | | University degree | Current student |
|---|---|---|---|---|---|---|---|---|---|---|
| | | | Male | Female | Other | Urban | Rural | Don't know | | |
| Argentina | 228 | 29.4 (8.9) | 151 (66.2%) | 75 (32.9%) | 2 (0.9%) | 214 (93.9%) | 10 (4.4%) | 4 (1.8%) | 63 (27.6%) | 103 (45.2%) |
| Australia | 449 | 36.7 (11.2) | 239 (53.2%) | 206 (45.9%) | 4 (0.9%) | 396 (88.2%) | 51 (11.4%) | 2 (0.4%) | 315 (70.2%) | 76 (16.9%) |
| Austria | 491 | 33.2 (11.7) | 210 (42.8%) | 275 (56.0%) | 6 (1.2%) | 320 (65.2%) | 168 (34.2%) | 3 (0.6%) | 281 (57.2%) | 176 (35.8%) |
| Brazil | 468 | 34.8 (13.2) | 178 (38.0%) | 286 (61.1%) | 4 (0.9%) | 449 (95.9%) | 17 (3.6%) | 2 (0.4%) | 363 (77.6%) | 140 (29.9%) |
| Canada | 399 | 35.9 (12.9) | 161 (40.4%) | 231 (57.9%) | 7 (1.8%) | 361 (90.5%) | 34 (8.5%) | 4 (1.0%) | 312 (78.2%) | 94 (23.6%) |
| China | 449 | 27.1 (9.5) | 208 (46.3%) | 239 (53.2%) | 2 (0.4%) | 361 (80.4%) | 85 (18.9%) | 3 (0.7%) | 300 (66.8%) | 249 (55.5%) |
| Egypt | 273 | 30.0 (10.9) | 118 (43.2%) | 155 (56.8%) | 0 (0.0%) | 250 (91.6%) | 13 (4.8%) | 10 (3.7%) | 263 (96.3%) | 65 (23.8%) |
| Georgia | 417 | 30.5 (10.3) | 89 (21.3%) | 327 (78.4%) | 1 (0.2%) | 385 (92.3%) | 28 (6.7%) | 4 (1.0%) | 373 (89.4%) | 120 (28.8%) |
| Germany | 634 | 31.0 (11.8) | 191 (30.1%) | 436 (68.8%) | 7 (1.1%) | 453 (71.5%) | 174 (27.4%) | 7 (1.1%) | 366 (57.7%) | 272 (42.9%) |
| India | 166 | 41.3 (17.1) | 119 (71.7%) | 45 (27.1%) | 2 (1.2%) | 162 (97.6%) | 4 (2.4%) | 0 (0.0%) | 138 (83.1%) | 35 (21.1%) |
| Indonesia | 395 | 37.8 (13.6) | 171 (43.3%) | 224 (56.7%) | 0 (0.0%) | 357 (90.4%) | 37 (9.4%) | 1 (0.3%) | 285 (72.2%) | 113 (28.6%) |
| Israel | 431 | 31.6 (12.4) | 190 (44.1%) | 238 (55.2%) | 3 (0.7%) | 364 (84.5%) | 58 (13.5%) | 9 (2.1%) | 186 (43.2%) | 187 (43.4%) |
| Italy | 434 | 31.0 (12.0) | 146 (33.6%) | 281 (64.7%) | 7 (1.6%) | 317 (73.0%) | 108 (24.9%) | 9 (2.1%) | 259 (59.7%) | 162 (37.3%) |
| Lebanon | 9 | 53.0 (12.5) | 2 (22.2%) | 7 (77.8%) | 0 (0.0%) | 8 (88.9%) | 1 (11.1%) | 0 (0.0%) | 9 (100.0%) | 1 (11.1%) |
| Maltese Islands | 470 | 38.2 (14.2) | 180 (38.3%) | 285 (60.6%) | 5 (1.1%) | 322 (68.5%) | 131 (27.9%) | 17 (3.6%) | 406 (86.4%) | 95 (20.2%) |
| Mexico | 401 | 30.2 (8.6) | 283 (70.6%) | 111 (27.7%) | 7 (1.7%) | 376 (93.8%) | 22 (5.5%) | 3 (0.7%) | 315 (78.6%) | 106 (26.4%) |
| the Netherlands | 430 | 31.9 (14.7) | 198 (46.0%) | 227 (52.8%) | 5 (1.2%) | 332 (77.2%) | 94 (21.9%) | 4 (0.9%) | 279 (64.9%) | 178 (41.4%) |
| Poland | 432 | 30.9 (11.7) | 145 (33.6%) | 278 (64.4%) | 9 (2.1%) | 368 (85.2%) | 59 (13.7%) | 5 (1.2%) | 288 (66.7%) | 159 (36.8%) |
| Portugal | 506 | 29.2 (10.9) | 142 (28.1%) | 360 (71.1%) | 4 (0.8%) | 396 (78.3%) | 102 (20.2%) | 8 (1.6%) | 365 (72.1%) | 258 (51.0%) |
| Serbia | 526 | 38.3 (13.3) | 128 (24.3%) | 397 (75.5%) | 1 (0.2%) | 466 (88.6%) | 51 (9.7%) | 9 (1.7%) | 351 (66.7%) | 98 (18.6%) |
| Singapore | 187 | 28.9 (11.6) | 87 (46.5%) | 99 (52.9%) | 1 (0.5%) | 171 (91.4%) | 5 (2.7%) | 11 (5.9%) | 96 (51.3%) | 91 (48.7%) |
| Slovenia | 458 | 31.2 (12.4) | 273 (59.6%) | 182 (39.7%) | 3 (0.7%) | 307 (67.0%) | 145 (31.7%) | 6 (1.3%) | 263 (57.4%) | 179 (39.1%) |
| Sweden | 518 | 41.3 (16.0) | 199 (38.4%) | 316 (61.0%) | 3 (0.6%) | 410 (79.2%) | 101 (19.5%) | 7 (1.4%) | 323 (62.4%) | 116 (22.4%) |
| United States | 362 | 32.5 (13.5) | 125 (34.5%) | 229 (63.3%) | 8 (2.2%) | 221 (61.0%) | 126 (34.8%) | 15 (4.1%) | 238 (65.7%) | 129 (35.6%) |
| Tunisia | 92 | 26.0 (8.9) | 40 (43.5%) | 52 (56.5%) | 0 (0.0%) | 87 (94.6%) | 4 (4.3%) | 1 (1.1%) | 55 (59.8%) | 53 (57.6%) |
| Türkiye | 480 | 40.0 (16.4) | 274 (57.1%) | 205 (42.7%) | 1 (0.2%) | 446 (92.9%) | 34 (7.1%) | 0 (0.0%) | 364 (75.8%) | 125 (26.0%) |
| United Kingdom | 422 | 38.7 (13.4) | 178 (42.2%) | 234 (55.5%) | 10 (2.4%) | 288 (68.2%) | 124 (29.4%) | 10 (2.4%) | 312 (73.9%) | 65 (15.4%) |
| Combined | 10,527 | 33.7 (13.3) | 4,425 (42.0%) | 6,000 (57.0%) | 102 (1.0%) | 8,587 (81.6%) | 1,786 (17.0%) | 154 (1.5%) | 7,168 (68.1%) | 3,445 (32.7%) |

demographic characteristics, including age, gender, university degree and political ideology (for further details, see Methods section 'Data analysis'). Instead of treating the one-item outcomes measured on seven-point scales (that is, climate change beliefs, worry and support for public action) as continuous[55], we apply cumulative probit regression models that appropriately treat the data as ordinal and can account for skewed response patterns[56]. We further specify informed hypotheses based on earlier research which allow us to test the presence versus absence of even small intervention effects[57]. All analyses are based on group differences in post-intervention outcomes. In sum, our analytical approach enables us to draw robust and valid conclusions about the data.

To simplify the interpretation and integration of the results into the existing literature, we supplement the informed Bayes factors with meta-analytic estimates of the overall intervention effect (Cohen's $d$) and the between-country heterogeneity ($\tau_c$) including confidence intervals (CI) derived from frequentist random-effects meta-analyses (Supplementary Information Section 2.1). As these estimates do not correspond to the specified models used for assessing the presence versus absence of intervention effects and can thus diverge from them, especially in the case of ordinal outcomes, the confidence intervals should not be interpreted as statistical significance tests. For future meta-analyses, we provide summary tables of the results for each outcome per country (Supplementary Table 1).

**Misperceptions of the reality consensus and crisis agreement**
In this section, we provide misperceptions of the reality consensus and crisis agreement per country before message exposure. These descriptives are unlikely to be representative of misperceptions per country due to the convenience sampling approach. Instead, they demonstrate that misperceptions are present in our samples—a prerequisite for consensus messaging to be effective.

Across all 27 countries ($n = 10,527$), the scientific consensus that human-caused climate change is happening (97%) is underestimated by, on average, −12.11% (95% CI (−12.43, −11.80)). This underestimation ranges from −20.91% (95% CI (−22.80, −19.03)) in the Chinese sample to −7.54% (95% CI (−8.53, −6.56)) in the German sample (Fig. 1). In total, 72.2% (95% CI (71.3, 73.0)) of participants underestimate this consensus, ranging from 57.5% (95% CI (52.2, 62.6)) in the US sample to 83.7% (95% CI (79.9, 87.0)) in the Chinese sample (Supplementary Information Section 2.2).

The scientific agreement that climate change constitutes a crisis (88%) is slightly underestimated by, on average, −4.14% (95% CI (−4.47, −3.81)). This ranges from an underestimation of −13.18% (95% CI (−15.07, −11.28)) in the Chinese sample to an overestimation of 1.18% (95% CI (0.14, 2.22)) in the German sample (Fig. 1). However, this scientific agreement is not consistently underestimated. A total of 44.5% participants across all countries (95% CI (43.6, 45.5)) underestimate the

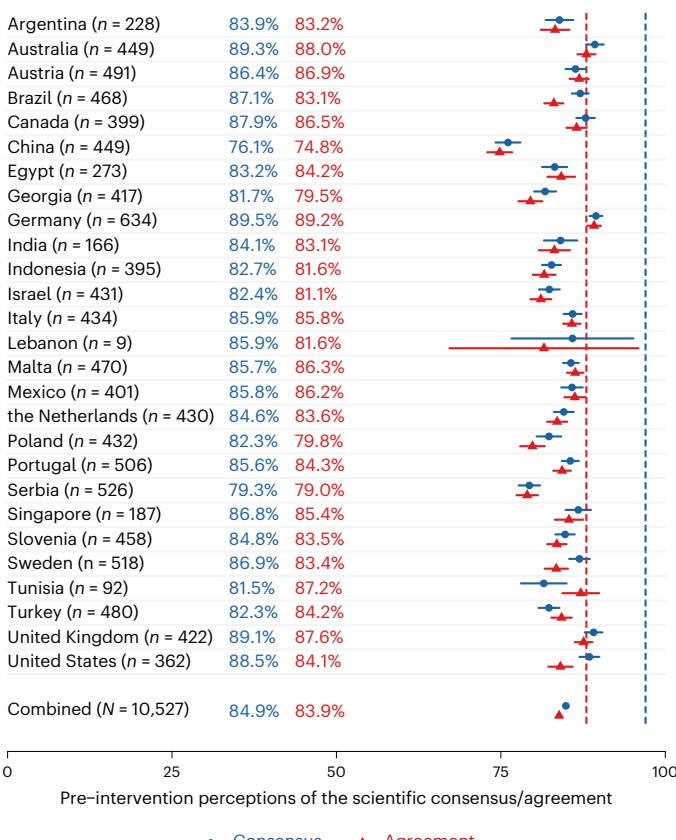

0 25 50 75 100

Pre–intervention perceptions of the scientific consensus/agreement

—●— Consensus —▲— Agreement

**Fig. 1 | Mean pre-intervention (mis)perceptions of the scientific consensus on the reality of climate change and agreement on climate change as a crisis per country sample.** The error bars represent the 95% CI for each country. The dashed blue line represents the actual scientific reality consensus (97%). The dashed red line represents the actual scientific crisis agreement (88%).

crisis agreement, ranging from 29.2% (95% CI (25.7, 32.9)) in the German sample to 69.5% (65.0, 73.7) in the Chinese sample (Supplementary Information Section 2.2).

Consistent with previous studies[17,19,20,24,25,58], we find substantial misperceptions of the reality consensus, now in a diverse 27-country sample, indicating a gap between the actual and the perceived scientific consensus that could be reduced with consensus messaging interventions. Although, on average, people also underestimate the crisis agreement, these misperceptions are relatively small.

### Effectiveness of the classic scientific consensus message
We first examine whether perceptions of the reality consensus, climate change beliefs (reality and human causation), worry and support for public action are higher in the classic scientific consensus ($n = 3,488$) compared to the control condition ($n = 3,512$).

Controlling for pre-intervention perceptions of the reality consensus, we find extremely strong support for H1$_a$ that post-intervention perceptions of the reality consensus are higher and thus more accurate in the classic scientific consensus compared to the control condition ($BF_{+0} = 2.01 \times 10^{12}$; Fig. 2). The Bayes factor implies that the data are $2.01 \times 10^{12}$ more likely under the hypothesis that participants in the classic consensus condition perceive the reality consensus as higher than those in the control condition (H$_+$) compared to the hypothesis that there is no difference between conditions (H$_0$). This corresponds to substantial effects between conditions across all countries, with Cohen's $d = 0.47$ (95% CI (0.41, 0.52)). However, these analyses also show extremely strong support for between-country heterogeneity

($BF_{10} = 1.49 \times 10^6$; $\tau_c = 0.06$, 95% CI (0.00, 0.15)), meaning that the effect on reality consensus perceptions is positive in all countries but varies across countries in terms of magnitude. Despite this strong evidence, the confidence interval includes zero and spans a wide range of values, indicating that the magnitude of heterogeneity is uncertain.

In line with H1$_b$ and H1$_c$, we find strong and extremely strong support that people believe more in climate change ($BF_{+0} = 25.51$) and human activity as its primary cause ($BF_{+0} = 467.86$; Fig. 2) after being exposed to the classic (H1$_c$, $n = 3,443$) compared to the control message (H1$_c$, $n = 3,464$). Both intervention effects are small (reality: Cohen's $d = 0.06$, 95% CI (0.01, 0.12); human causation: Cohen's $d = 0.10$, 95% CI (0.04, 0.15)), with evidence against any between-country heterogeneity (reality: $BF_{10} = 4.78 \times 10^{-6}$, $\tau_c = 0.07$, 95% CI (0, 0.20); human causation: $BF_{10} = 1.51 \times 10^{-5}$, $\tau_c = 0.06$, 95% CI (0, 0.17)).

Similarly, there is moderate support for a small but consistent effect of the classic scientific consensus intervention on climate change worry (H1$_d$; $BF_{+0} = 5.03$; Cohen's $d = 0.05$, 95% CI (−0.01, 0.10); Fig. 2), with evidence against between-country heterogeneity ($BF_{10} = 9.93 \times 10^{-10}$, $\tau_c = 0.07$, 95% CI (0, 0.21)). In contrast to H1$_e$, we find weak evidence against an effect of the classic scientific consensus message on support for public action ($BF_{+0} = 0.62$; Cohen's $d = 0.02$, 95% CI (−0.03, 0.08); Fig. 2), with evidence against between-country heterogeneity ($BF_{10} = 2.93 \times 10^{-10}$, $\tau_c = 0.06$, 95% CI (0, 0.20)).

Additionally, we explore whether the classic message influences belief in climate change as a crisis (not preregistered). We find strong evidence that the classic scientific consensus message increases belief in climate change as a crisis ($BF_{+0} = 35.80$, Cohen's $d = 0.06$, 95% CI (0.01, 0.11)), with evidence against between-country heterogeneity ($BF_{10} = 5.18 \times 10^{-10}$, $\tau_c = 0$, 95% CI (0, 0.12)).

### Effectiveness of the updated scientific consensus message
We next compare the updated consensus ($n = 3,527$) to the control ($n = 3,512$) and the classic consensus condition ($n = 3,488$). Controlling for pre-intervention perceptions of the reality consensus and crisis agreement, respectively, we find extremely strong support for H2$_a$ and H2$_b$ that perceptions of both the reality consensus ($BF_{+0} = 2.12 \times 10^{12}$; Cohen's $d = 0.47$; 95% CI (0.41, 0.52); Fig. 2) and the crisis agreement ($BF_{+0} = 1.54 \times 10^5$; Cohen's $d = 0.23$; 95% CI (0.16, 0.31); Fig. 2) are higher in the updated compared to the control condition, with substantial evidence for relatively small between-country heterogeneity (reality consensus: $BF_{10} = 1.19 \times 10^3$, $\tau_c = 0.05$, 95% CI (0, 0.14); crisis agreement: $BF_{10} = 1.53 \times 10^8$, $\tau_c = 0.15$, 95% CI (0.09, 0.23)). For climate change beliefs (H2$_c$ and H2$_d$), worry (H2$_f$) and support for public action (H2$_g$), the effects of the updated condition are consistent with those of the classic condition in terms of evidence strength and effect size (Fig. 2 and Supplementary Information Section 2.3). Contrary to H2$_e$, there is only weak support for an effect of the updated message on belief in climate change as a crisis ($BF_{+0} = 1.80$; Cohen's $d = 0.04$, 95% CI (−0.01, 0.09); Fig. 2), with evidence against between-country heterogeneity ($BF_{10} = 9.15 \times 10^{-09}$, $\tau_c = 0.06$, 95% CI (0, 0.14)).

Comparing the updated and classic consensus condition (H3$_{a–c}$) reveals consistent moderate-to-strong evidence for no added benefit of the updated message in terms of crisis belief, worry and support for public action (crisis belief: $BF_{+0} = 0.09$, Cohen's $d = −0.02$, 95% CI (−0.07, 0.03); worry: $BF_{+0} = 0.11$, Cohen's $d = −0.01$, 95% CI (−0.06, 0.04); public action: $BF_{+0} = 0.25$, Cohen's $d = 0$, 95% CI (−0.04, 0.05)), all with extremely strong evidence against any heterogeneity across countries (crisis belief: $BF_{10} = 1.72 \times 10^{-8}$, $\tau_c = 0.07$, 95% CI (0, 0.14); worry: $BF_{10} = 9.84 \times 10^{-11}$, $\tau_c = 0.06$, 95% CI (0, 0.17); public action: $BF_{10} = 3.82 \times 10^{-11}$, $\tau_c = 0$, 95% CI (0, 0.08)). We also find extremely strong exploratory evidence that participants are more confident in their agreement perceptions after seeing the updated compared to the classic consensus message (not preregistered; $BF_{10} = 4.90 \times 10^8$; Cohen's $d = 0.44$; 95% CI (0.38, 0.51); between-country heterogeneity: $BF_{10} = 73.60$, $\tau_c = 0.10$, 95% CI (0.02, 0.19)).

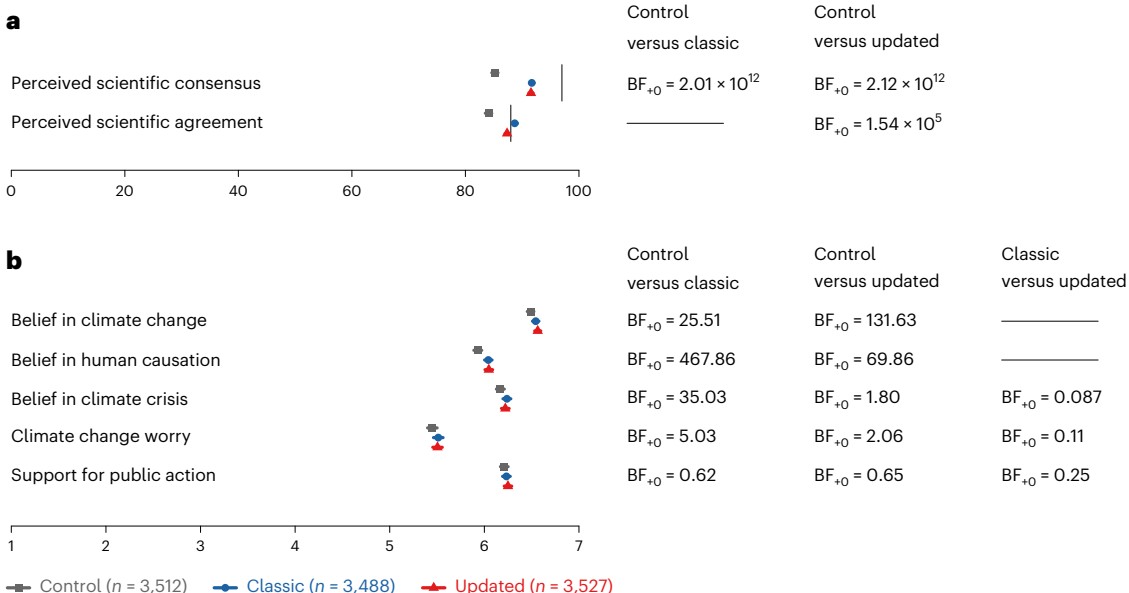

**Fig. 2 | Effects of the classic and updated scientific consensus intervention on all post-intervention outcomes. a**, shows that both the classic and the updated consensus messages increase perceived scientific consensus and agreement. The upper vertical line represents the actual scientific reality consensus (97%); the lower vertical line represents the actual scientific crisis agreement (88%). **b**, shows that both messages increase belief in climate change, its human causation and climate change worry but not support for public action. The updated condition does not further increase belief in crisis, worry or support for public action. **a**,**b**, the means of each outcome per condition and the 95% CI (which are too small to be visible in panel **a**) are presented on the left-hand side. On the right-hand side, Bayes factors for between-group comparisons are shown. We only indicate Bayes factors for the tested hypotheses, not all comparisons.

'Classic' refers to the message communicating the scientific consensus on the reality of climate change. 'Updated' refers to the message communicating the scientific consensus on the reality of climate change and the scientific agreement on climate change as a crisis. Across both panels, sample sizes for all outcomes are: $n_{classic} = 3,488$; $n_{updated} = 3,527$; and $n_{control} = 3,512$, except for belief in human causation of climate change, where: $n_{classic} = 3,443$; $n_{updated} = 3,490$; and $n_{control} = 3,464$. These results are reported in the sections 'Misperceptions of the reality consensus and crisis agreement', 'Effectiveness of the updated scientific consensus message' and 'Effectiveness of the updated scientific consensus message'. Complete results are described in the Results and the Supplementary Results.

## Moderators of consensus messaging effects on misperceptions

We next investigate whether message familiarity, trust in climate scientists and political ideology moderate (1) the effectiveness of the classic consensus message on reality consensus perceptions and (2) the effectiveness of the updated consensus message on perceptions of the reality consensus and crisis agreement, compared to the control condition. In all analyses, we control for pre-intervention perceptions of the reality consensus and/or crisis agreement, depending on the respective outcome. We do not estimate cross-country heterogeneity for any of the moderation effects due to limited statistical power for these tests.

In line with H4$_a$, there is extremely strong evidence (BF$_{10}$ = $1.43 \times 10^{16}$) that the classic consensus message is more effective at correcting misperceptions for those who reported being less familiar with the message before the study. Contrary to H4$_b$, we find extremely strong evidence (BF$_{10}$ = 0) against the assumption that the classic consensus intervention is more effective for people who trust climate scientists more. Further unplanned exploratory analyses suggest a three-way interaction. Those who trust climate scientists more have higher and more accurate perceptions of the scientific consensus before the intervention ($r = 0.20$), leaving relatively less room for updating beliefs in the experimental conditions (Supplementary Information Section 2.4; Fig. 1a). Finally, we find weak evidence (BF$_{10}$ = 1.89) for a moderating effect of political ideology. In contrast to H4$_c$, the intervention seems slightly more effective for people on the right of the political spectrum. As for trust in climate scientists, this moderation probably also depends on the higher degree of possible improvement among right- compared to left-leaning participants (Supplementary Information Section 2.4; Fig. 1b).

Comparing the updated and control messages (Q5), there is extremely strong support for a moderation by message familiarity, trust in climate scientists and political ideology. These moderations are consistent with the ones observed for the effectiveness of the classic intervention, such that people who are less familiar with the message (reality consensus: BF$_{10}$ = $2.53 \times 10^{22}$; crisis agreement: BF$_{10}$ = $2.76 \times 10^{19}$), trust climate scientists less (reality consensus: BF$_{10}$ = $3.34 \times 10^{50}$; crisis agreement: BF$_{10}$ = $2.60 \times 10^{90}$) and are more right-leaning (reality consensus: BF$_{10}$ = $2.52 \times 10^{5}$; crisis agreement: BF$_{10}$ = $3.10 \times 10^{16}$) update their perceptions of the reality consensus and crisis agreement more. Similar to the interactions with trust in climate scientists and political ideology for the classic consensus message, additional exploratory analyses suggest that these moderations are qualified by pre-intervention perceptions of the reality consensus and crisis agreement. People with more right-leaning ideology and lower trust in climate scientists had lower initial consensus/agreement perceptions and thus a wider margin to update (Supplementary Information Sections 2.4; Fig. 1c,d).

## Additional exploratory analyses

As specified in the preregistration, we run several exploratory analyses. Rerunning all main confirmatory analyses without demographic control variables yields highly similar results (Supplementary Information Section 2.5). Further moderation analyses show extremely strong evidence that the classic compared to the control message is more effective at correcting misperceptions of the reality consensus for people with lower (that is, more incorrect) pre-intervention perceptions of this consensus (BF$_{10}$ = $6.87 \times 10^{211}$). We also find strong evidence for a similar moderation on belief in the reality of climate change—the effect of the classic consensus (versus control) message was larger for those with lower pre-intervention perceptions of the reality consensus,

such that they increased their belief in the reality of climate change to a greater extent ($BF_{10} = 12.47$). We find weak evidence against the same moderation for belief in the human causation of climate change ($BF_{10} = 0.69$), climate change worry ($BF_{10} = 0.89$) and support for public action ($BF_{10} = 0.78$).

Following a reviewer's suggestion, we also explore whether country-level characteristics, such as individualism–collectivism and power distance, moderate the effects of both interventions on perceptions of the reality consensus and crisis agreement. We find no convincing evidence for any moderation by these two cultural dimensions (Supplementary Information Section 2.6). However, we detect only weak evidence against or for any country-level moderation effects, which suggests that this study is underpowered to robustly probe such moderations. These results should, therefore, be seen as tentative and followed up by analyses on datasets including more countries.

## Discussion

Across 27 countries on six continents, we test the effectiveness of two climate science consensus messages, a classic message on the reality of climate change and an updated message that additionally emphasizes the agreement among scientists on climate change as a crisis. We find substantial misperceptions of the scientific consensus that human-caused climate change is happening across all country samples—indicating a gap between the actual and the perceived scientific consensus that could be reduced with consensus messaging interventions. However, perceptions of the crisis agreement are relatively accurate in all country samples.

Complementing previous studies that relied primarily on US samples[26,32,33], informing people about the 97% scientific consensus on the reality of climate change is largely effective in a more diverse 27-country sample. Specifically, the classic scientific consensus intervention substantially increases perceptions of the scientific consensus (Cohen's $d = 0.47$, 95% CI (0.41, 0.52)) as well as—to a smaller extent—beliefs in the reality ($d = 0.06$, 95% CI (0.01, 0.12)) and human causation ($d = 0.10$, 95% CI (0.04, 0.15)) of climate change, as well as worry ($d = 0.05$, 95% CI (−0.01, 0.10)). However, we find weak evidence ($BF_{+0} = 0.62$) against a direct effect on support for public action ($d = 0.02$, 95% CI (−0.03, 0.08)). The magnitude of the effects is generally in line with recent meta-analyses which found moderate effects on revising consensus perceptions and small effects on outcomes that are more remote from the intervention, such as climate change beliefs and worry[26]. As the conceptually most remote outcome from the intervention, it is perhaps unsurprising that this study finds no direct effect on support for public action, although some studies, conducted predominantly in the United States, report significant indirect[13,25] and direct effects[24]. In sum, while consensus messaging on climate change can produce small shifts in personal beliefs and worry on the topic of climate change, a one-time messaging intervention alone seems insufficient to alter preferences toward major policy topics, which has also been noted in previous meta-analytic work[26,27].

The updated consensus message shows similar effects as the classic message on beliefs in the reality ($d = 0.07$, 95% CI (0.02, 0.12)) and human causation ($d = 0.09$, 95% CI (0.04, 0.15)) of climate change, as well as worry ($d = 0.04$, 95% CI (−0.01, 0.08)), with weak evidence ($BF_{+0} = 0.65$) against a direct effect on support for public action ($d = 0.02$, 95% CI (−0.02, 0.07)). We find only weak evidence for a positive effect on crisis belief ($BF_{+0} = 1.80$; $d = 0.04$, 95% CI (−0.01, 0.09)). Comparing the updated to the classic message showed no added value in additionally communicating the 88% scientific agreement on climate change as a crisis beyond strengthening confidence in perceptions of the crisis agreement. These findings might be due to participants' perceptions of the scientific agreement being already relatively accurate before message exposure. Additionally, given that even small perceived dissent among environmental scientists can undermine message effectiveness[15,59,60], the tested scientific agreement (88%) might not be high

enough, and consequently convincing enough, to further shift belief in climate change as a crisis, worry and support for public action. Therefore, more attention needs to be devoted to effective ways of communicating the very high scientific confidence about the adverse consequences of climate change and the urgency of climate action to curb these impacts expressed in the Sixth IPCC report[35].

The results of the current study also provide several useful indicators for selecting target audiences for consensus messaging interventions. Consistent with Bayesian models of information processing[39–41], the message seems to be more effective for individuals who report being less familiar with it before exposure. Our exploratory analyses suggest that people with lower initial perceptions of the consensus increase their estimates to a greater extent, probably because they have more 'room' to update their perceptions. They also increase their belief in the reality of climate change more than people with higher initial perceptions. Consequently, repeated exposure to scientific consensus messages might have diminishing returns. As the effect on consensus perceptions is detectable days and weeks after exposure[31,61,62], people might become more accurate and familiar with the message as they are repeatedly exposed to it, which, in turn, would yield increasingly smaller effects. While this decay in effectiveness is probable when people are not exposed to contrarian views, the information ecosystem contains climate misinformation and disinformation[63,64], particularly in contexts where climate change is a politicized topic. In such contexts, consensus messaging can neutralize counterarguments[23] and repeated consensus message exposure is effective for those who report being exposed to a mix of contradicting information between two exposures[61].

Contrary to some motivated cognition accounts and findings[43,44], consensus messaging does not seem to backfire for people whose worldviews might not align with the scientific consensus on climate change, such as right-leaning individuals or those with lower trust in climate scientists. Rather, the present study supports previous research that found consensus messages to result in larger belief updating for those with right-leaning political ideologies[24,31] and extends this to those with lower trust in climate scientists, as these groups tend to have higher initial misperceptions. This means that a left-leaning person with a pre-intervention consensus estimate of 75% is likely to update their consensus perceptions more than a right-leaning person with the same consensus estimate. However, at the group level, targeting low-trust and right-leaning individuals corrects misperceptions to a greater extent. As people across the ideological and trust spectrums still update their estimates, consensus messaging represents a non-polarizing tool useful for reaching a social consensus on climate change across different audiences.

While the present study tested the practical use and general effectiveness of scientific consensus messages across countries, we recognize several limitations. First, as we focus on the direct effects of the interventions on several outcomes, this work does not speak to theoretical predictions of the Gateway Belief Model[24,33]—the main theoretical framework for scientific consensus messaging—that focuses on cascading indirect effects of consensus messaging through changes in perceived consensus and further through climate change beliefs to support for public action. We did not measure pre-intervention estimates for all beliefs, which precludes formal modelling of the 'gateway' process.

Second, this study finds nominally small effects[65] of scientific consensus messaging on personal climate change beliefs and worry. However, these effects are in line with previous research[26,27] and can be practically relevant[66], as the intervention is easily scalable to reach many people because of its brevity. Targeting specific subgroups, such as those on the political right who are most likely to underestimate the consensus, might also increase its overall effectiveness.

Third, social-media users are generally younger, more educated, more liberal, more likely to be female and pay more attention

to politics[67,68], which is also reflected in our current samples. On the one hand, our social-media-based sampling approach may have led us to underestimate the intervention effects because, for example, younger and more educated individuals are more likely to believe in climate change[69], which is, in turn, associated with higher perceptions of the scientific consensus[70]. On the other hand, several previous studies have shown that average treatment effects can be accurately estimated in experiments using convenience samples[71,72]. This is also supported by the fact that the effect size estimates for the effectiveness of the classic message in our study (misperception correction, $d = 0.47$; climate change attitudes, $d = 0.05–0.10$) relatively closely align with effect size estimates from previous meta-analyses on scientific consensus messaging (misperception correction, $g = 0.56$; climate change attitudes, $g = 0.09–0.12$) that rely on mostly US-based studies with nationally representative samples. From a practical perspective, hard-to-reach populations (for example, people who do not have access to the internet or do not use social media) will probably not be exposed to and thus influenced by a scientific consensus message when used by policy-makers in, for example, online campaigns. We do not discount the importance of those populations; we simply highlight this consideration in the context of the effectiveness of this specific intervention.

Fourth, we are unable to draw definitive conclusions about the extent of between-country heterogeneity and make concrete recommendations as to where scientific consensus messaging might be most effective, due to the convenience sampling approach and insufficient statistical power to detect moderations by country-level predictors (that is, cultural dimensions). We encourage future research to continue testing message effectiveness within countries using representative samples and, possibly, our materials and translations, to ultimately make practical recommendations for climate change communication tailored to specific countries (for example, as previously done in Germany[73]). In addition, datasets including many countries are essential for robustly testing country-level factors that might determine consensus messaging effectiveness.

Last, we recognize the limitations of single items, especially for broader constructs, such as support for public action[24]. Future research might investigate the effects of scientific consensus messages on specific climate change mitigation policies. As beliefs in climate change and human causation may not only be associated with support for mitigation but also adaptation policies[74], we encourage further studies to investigate the effectiveness of scientific consensus messaging on climate change adaptation policies[32]. Effects on climate change mitigation are especially impactful in countries with high carbon emissions[32], whereas climate change adaptation might prove more useful in nations with comparably lower carbon emissions that are, at the same time, disproportionately affected by climate change[75].

## Conclusion

Across more than 10,000 participants and 27 countries, this study shows that scientific consensus messages on climate change can reduce consensus misperceptions and produce small shifts in climate change beliefs and worry. This effect does not directly extend to support for public action. Communicating the scientific agreement that climate change is a crisis, along with the consensus that human-caused climate change is happening, seems to have no added value beyond strengthening confidence in perceptions of the crisis agreement. This underscores the importance of continuing to investigate effective ways to communicate climate science projections, beyond the consensus that human-caused climate change is happening. Crucially, scientific consensus messages are most effective among people who were less familiar with the message and had less accurate initial consensus perceptions, including those with lower trust in climate scientists and right-leaning political ideologies. In sum, scientific consensus messaging is an effective, non-polarizing tool for substantially reducing scientific consensus misperceptions

and slightly shifting personal climate change beliefs and worry across samples and audiences.

## Methods
### Ethics
We obtained ethical approval from the Institutional Review Board from the University of Amsterdam (the Netherlands; protocol FMG-1123) and the University of Porto (Portugal; protocol 2023/06-12). All participants provided informed consent at the beginning of the survey experiment. Participation was voluntary and not compensated. However, in Canada and Mexico, we supplemented the unpaid samples with paid participants using Prolific. We collected paid samples in countries in which (1) we could not achieve the target sample size through convenience sampling and (2) enough participants were available on Prolific.

### Participant recruitment
We recruited participants using an existing network of researchers that used scalable methods to collect large, diverse samples in 27 countries (Argentina, Australia, Austria, Brazil, Canada, China, Egypt, Georgia, Germany, India, Indonesia, Israel, Italy, Lebanon, the Maltese Islands, Mexico, the Netherlands, Poland, Portugal, Serbia, Singapore, Slovenia, Sweden, Tunisia, Türkiye, the United Kingdom and the United States) from 27 July to 4 August 2023. The final set of countries was selected according to our collaborators' familiarity and connections with the countries, aiming for geographic spread with at least one country per continent, except Antarctica[76].

We collected convenience sampling using snowballing, mailing lists, social media and Prolific (only in Canada and Mexico). On social media, we posted in special interest groups that relate to current events, popular culture or media discussions. We also posted comments on discussion threads of major news stories unrelated to climate change or sustainability. These approaches have been effective at recruiting a diverse body of participants in similar research where comparable sample sizes were required[76,77]. Crucially, because climate change is a prominent topic in public discourse, we advertised the study as a survey on popular media topics, to prevent selection bias of participants with strong opinions on climate change.

### Sample details
To be able to participate in this study, participants needed to be at least 18 years old, live in one of the 27 target countries and speak the language in which the survey was conducted (that is, the most prominent locally spoken language/s) fluently. Of 21,462 individuals who clicked on the link, 11,702 participants completed the study, while 676 were filtered out at the beginning of the survey because they did not reside in any of the 27 countries. Out of all people who dropped out, most did so after seeing the informed consent (2,687; 29.6%), after the introduction that they are randomly assigned to one topic but before seeing that this topic is climate change (804; 8.8%) and right after the control/intervention message (931; 10.2%). Consensus and agreement perceptions of individuals in the two intervention conditions who dropped out directly after seeing the message (consensus, 78.4% and 79.1% in the classic and updated condition; agreement, 75.3% and 77.1%) are slightly lower compared to perceptions of those who completed the study (consensus, 84.2% and 84.5%; agreement, 83.4% and 83.3%), indicating selective dropout. However, the dropout rate (that is, number of dropouts directly after seeing the control/intervention message versus number of overall dropouts) is comparable between both intervention (classic: 267, 2.9%; updated: 296, 3.3%) and control conditions (368, 4.1%). This indicates that the dropout is unlikely to result from a specific backfire of the consensus messages and is suggestive of a more general tendency of less motivated participants to trickle out of the survey in its initial stages. After data exclusions (see section 'Data analysis'), 10,527 participants (including Canada, $n_{paid} = 179$ and Mexico, $n_{paid} = 143$) remained for the analysis.

## Design and procedure

We conducted an online study using Qualtrics. The study implemented a between-participants design with three conditions—consensus, updated consensus and control. As the only exception to this, we measured reality consensus and crisis agreement perceptions both before and after the intervention (including masking the study aims and distractors to minimize demand effects). This is because this 'estimate and reveal' technique makes the intervention more effective[20], as it highlights the gap between the participants' perceptions and the scientific norm[33,61]. Therefore, we measured pre-intervention reality consensus and crisis agreement perceptions and control for them in estimating between-participants effects of the two interventions on post-intervention consensus perceptions. The outline of the procedure is depicted in Supplementary Fig. 2. The median completion time was 6.33 min.

**Pre-intervention.** As part of the informed consent, the topic of the research was described as 'opinions about and reactions to popular news topics' to reduce potential self-selection biases as well as biasing participants' responses. After providing informed consent, we asked for participants' current country of residence. If they did not live in any of the targeted countries, they were redirected to the end of the survey. The remaining participants were told that they were now asked to provide their opinion on one out of 20 randomly selected news topics. In fact, all participants answered questions about climate change.

Next, to ensure equal understanding of the topic across testing contexts, all participants saw a brief description of climate change as a news topic ('You may have noticed that climate change has been getting some attention in the news. Climate change refers to the idea that the world's average temperature has been increasing over the past 150 years, may be increasing more in the future and that other aspects of the world's climate may change as a result') used in previous research[78].

Participants then responded to two items assessing climate change consensus perceptions in a randomized order (see section 'Materials' for item wording).

**Intervention.** Following previous studies on climate change consensus messaging[44], participants were informed that they would see a random statement from a large database of media statements the researchers maintain and were randomly, double-blind assigned to one of the three experimental conditions (consensus, updated consensus or control). Participants in the consensus condition saw the classic message on the scientific consensus emphasizing the reality of climate change: '97% of climate scientists agree that human-caused climate change is happening'. Participants in the updated consensus condition were shown the classic consensus message, supplemented with the IPCC scientists' agreement that climate change is a crisis: '97% of climate scientists agree that human-caused climate change is happening. In addition, 88% of climate scientists agree that climate change constitutes a crisis'. We distinguish between reality consensus and crisis agreement to emphasize that scientific consensus on the reality of climate change was obtained by analysing abstracts of scientific publications, while the scientific agreement with regards to climate change as a crisis was obtained by surveying IPCC authors (that is, percentage of IPCC authors who agree with the statement that climate change is a crisis). See Supplementary Information Sections 3.3 and 3.4 for further information on the wording choice and the pilot study. To reduce anchoring effects, the control group was shown an unrelated consensus message: '97% of dentists recommend brushing your teeth twice per day'[61].

**Post-intervention.** Similar to previous studies on consensus messaging[24,61], as a distractor task, all participants read a paragraph about an upcoming science fiction film, Dune 2, and were asked one filler question about the movie. They also responded to an attention check.

After, they again reported their perceptions of the scientific consensus on the reality of climate change and agreement that climate change is a crisis (randomized and consistent with pre-intervention order for each participant), their confidence in these estimates directly after each one, as well as their personal beliefs in climate change beliefs, worry and support for public action (see following section 'Materials'). After reporting demographic information, they answered several questions that tap into potential moderators (message familiarity, trust in scientists and political ideology). Finally, they completed a comprehension check and were debriefed.

## Materials

**Outcomes.** *Perceived scientific consensus on the reality of human-caused climate change.* 'To the best of your knowledge, what percentage of climate scientists agree that human-caused climate change is happening?'. Participants responded on a slider scale from 0% to 100%. See ref. 24.

*Perceived scientific agreement about climate change as a crisis.* 'To the best of your knowledge, what percentage of climate scientists agree that climate change constitutes a crisis?' Response options again ranged from 0% to 100%.

*Confidence in both scientific consensus and agreement estimates (exploratory).* 'How certain are you about your answer above?'. Participants indicated the confidence in their estimates on a scale from 0 (very uncertain) to 100 (very certain). See ref. 20.

*Belief in climate change.* 'How strongly do you believe that climate change is or is not happening?'. Responses were provided on a scale from 1 (I strongly believe climate change is not happening) to 7 (I strongly believe climate change IS happening). See ref. 24.

*Belief in the human causation of climate change.* 'How much of climate change do you believe is caused by human activities, natural changes in the environment or a combination of both?'. Response options ranged from 1 (I believe climate change is caused mostly by natural changes in the environment) to 7 (I believe climate change is caused mostly by human activities). Participants also had the option to report that they believe climate change is not happening. Adapted from ref. 24.

*Belief in climate change as a crisis.* 'How strongly do you believe that climate change constitutes a crisis?'. Response options ranged from 1 (I strongly believe climate change is not at all a crisis) to 7 (I strongly believe climate change is a crisis).

*Climate change worry.* 'How worried are you about climate change?' Response options ranged from 1 (I am not at all worried about climate change) to 7 (I am very worried about climate change). See ref. 24.

*Support for public action on climate change.* 'Do you think society should be doing more or less to reduce climate change?'. Response options ranged from 1 (much less), 4 (same amount) to 7 (much more). Adapted from ref. 24.

**Checks.** *Filler item.* 'How likely are you to watch Dune 2?', including the response options 1 (very unlikely) to 5 (very likely).

*Attention check.* 'This is a test item. Please select 'somewhat agree''. The response scale included 1 (strongly disagree), 2 (somewhat disagree), 3 (neither agree nor disagree), 4 (somewhat agree) and 5 (strongly agree).

*Comprehension check.* 'As part of this survey, you may have viewed one of the statements below. Please select the statement, if any, you have seen'. Participants could select one of four options: the control, classic consensus or updated consensus message as well as none of the above.

**Demographic information.** We assessed current country of residence ('Do you currently live in (country)?'), age ('What is your age in years?'), gender ('What is your gender?', including male, female, other and prefer not to say as response options), current region of residence ('In which region do you currently live?'; response categories were adapted to each country), urbanicity ('Would you describe the area where you live as urban or rural?', including urban, rural and don't know), highest education level ('What is the highest level of education you have received?', including seven levels, from 1 less than high school to 7 doctoral degree), student status ('Are you currently a college/university student?', yes/no) and ethnicity ('Please choose which best describes you'; response categories were adapted to the country-specific context). While age, gender, education and political ideology were also used as covariates in confirmatory analyses, the remaining demographic information was used to describe the samples.

**Moderators.** *Familiarity with the consensus/agreement statements.* We assessed participants' familiarity with the consensus messages using two items: (1) 'Before taking this survey, to what extent were you familiar with the following statement: '97% of climate scientists agree that human-caused climate change is happening'?' and (2) 'Before taking this survey, to what extent were you familiar with the following statement: '88% of climate scientists agree climate change constitutes a crisis'?'. Response options for both questions ranged from 1 (not at all familiar) to 7 (very familiar). Adapted from ref. 42.

*Trust in climate scientists.* 'In general, how much do you trust or distrust climate scientists as a source of information about climate change?' Response options ranged from 1 (strongly distrust) to 7 (strongly trust). Adapted from ref. 61.

*Political Ideology.* We assessed ideology using one item: 'In politics, people sometimes talk of 'left' and 'right'. Where would you place yourself on this scale, where 0 means the left and 10 means the right?' from 0 (left) to 10 (right).

**Translation.** We used a standard forward-back translation procedure for the instrument, following the Psychological Science Accelerator's[79] recommendations. Specifically, the instrument was first translated into the local language by at least one researcher who is fluent in English and the local language. Another independent researcher, also fluent in both languages, then back-translated this version to English, which was compared to the original English instrument. The translators resolved any disagreements through discussion. After the wording had been agreed upon, independent team members performed fidelity checks for each Qualtrics survey to ensure equal formatting and verify the survey flow. The final version was pretested in the target population ($n \approx 5$) for clarity checks. All necessary cultural adjustments are reported in Supplementary Information Section 3.5.

**Data analysis**
**Preprocessing.** *Exclusions.* All countries were included in the analyses, regardless of the achieved sample size. At the participant level, we excluded respondents who failed the attention check ($n = 1,032$) or finished the survey in <2 min ($n = 0$; based on pilot data where none of the respondents completed a similar survey in <2 min). In addition, for paid samples, responses identified as potential bots (Qualtrics variable: Q_RecaptchaScore < 0.50; $n = 3$), duplicates (Qualtrics variable: Q_RelevantIDDuplicateScore ≥ 75; $n = 9$) or fraudulent (Qualtrics variable: Q_RelevantIDFraudScore ≥ 30, as per Qualtrics recommendations; $n = 6$) were excluded. Given that these are not equally functional and thus informative across countries, we kept respondents for whom these metrics were not recorded. All exclusion criteria were preregistered.

*Missing data.* The following responses were coded as missing values: 'prefer not to answer/say' (items: gender and ethnicity) and 'I believe that climate change is not happening' (item: human causation of climate change). Participants with missing data were excluded on an analysis-by-analysis basis. Before data collection, we expected few missing values due to the forced-response format of all items. This was confirmed, with 130 missing values on belief in the human causation of climate change and 128 missing values on gender.

*Outliers.* We did not define or remove any outliers, as all measures are bounded, which effectively prevents any outliers.

**Main analysis.** We analysed the data using the Bayesian model-averaging framework[53,54,80] with mixed-effects models. This allowed us to evaluate the evidence in favour and against the preregistered hypotheses[81,82] while accounting for uncertainty in the model structure (for example, constant versus heterogeneous intervention effects across countries). For hypotheses about the continuous outcomes (perceptions of the reality consensus and crisis agreement), we ran mixed-effects linear regression models estimated using the BayesFactor[83] R package and 100,000 iterations. For hypotheses about the ordinal outcomes (personal belief in climate change, human causation and climate change as a crisis, climate change worry and support for public action), we used mixed-effects cumulative probit regression models in Stan[84] and through the Rstan R package[85] and computed the marginal likelihood by means of bridge sampling using the bridgesampling R package[86]. These models were run with two chains, of which each included 2,000 warm-up and 3,000 sampling iterations. All models converged with $\hat{R} < 1.02$.

The cumulative probit regression models allowed us to deal with the skewed responses and different ordinal scale response patterns (no, random, constant and dominant)[56].

In all models, we accounted for nesting of participants (level 1) within countries (level 2). In all analyses, we model-averaged across models assuming the presence versus absence of random-slopes of intervention (that is, differences in the intervention effect across countries) and adjusted for demographic covariates (age, gender, university degree and political ideology), unless stated otherwise. We supplemented the interpretation of Bayes factors with the evidence labels of ref. 87 based on ref. 88.

*Previous settings.* In the linear mixed-effects models, we set a prior scale of $r = 0.50$ for the fixed-effects regression coefficients, the 'medium' prior scale closely corresponding to the previously reported meta-analytic effect of $g = 0.55$ (ref. 26) and $r = 0.25$ for the random-effects regression coefficients, assuming that the between-country variability is approximately half the effect size. The common intercept and residual variance use the default Jeffreys prior.

In cumulative probit mixed-effect models, we set the prior standard deviation on the latent scale shift to $\theta = 0.14$ (which converts to the Cohen's $d$ of approximately 0.10 were the data analysed as continuous which corresponds to recent meta-analyses[26,27]) and the standard deviation of the normal distribution for the random effects to $\tau = 0.07$, again assuming that the between-country heterogeneity is approximately half the effect size. The common thresholds use the default standard normal prior distribution[56].

The Supplementary Information Section 3.6 contain a detailed overview of our analytical approach and the Bayes factor design analyses. Detailed model specifications are available on OSF (https://osf.io/z6quh/).

**Deviations from the preregistration**
We preregistered this study on OSF before data collection (https://osf.io/b6cmp; 19 July 2023). We deviated from the preregistered plan in the following ways. First, we placed the comprehension check after,

instead of before (as preregistered), the demographics and moderator sections to avoid compromising moderator validity. More specifically, if we had followed our preregistered item order, the message familiarity measure would have been less valid because the participants would have already seen all messages in the comprehension check question. Participants were exposed to one of the three messages, indicated their familiarity with both consensus messages and then reported which of the messages they had seen. Since all participants have seen both messages as part of the familiarity measure, we did not use the comprehension check in any of our analyses.

Second, we did not measure subjective income as specified in the preregistration, to keep the survey brief and because we did not plan to use it in the main analyses (unlike, for example, education). Both decisions were made before preregistering and we missed revising the protocol before preregistering. In addition, there was a typo in the preregistration about exclusion criteria: for paid samples, we excluded participants with the Q_RelevantIDFraudScore ≥ 30, instead of <30 as stated erroneously in the preregistration. This is in line with official Qualtrics guidelines on using this fraud indicator (https://www.qualtrics.com/support/survey-platform/survey-module/surveychecker/fraud-detection).

### Reporting summary

Further information on research design is available in the Nature Portfolio Reporting Summary linked to this article.

## Data availability

The raw and cleaned datasets for all analyses reported in this manuscript are publicly available under a CC-By Attribution International 4.0 license on the OSF (https://osf.io/z6quh/).

## Code availability

All survey materials as well as R code for the main analyses and the Bayesian Design analyses are publicly available on the OSF (https://osf.io/z6quh/).

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

## Acknowledgements

We would like to thank the Junior Researcher Programme, the Global Behavioral Science (GLOBES) coordinators from Undergraduate Global Engagement at Columbia University and the Centre for Business Research in the Judge Business School as well as Corpus Christi College at the University of Cambridge. We also thank the Rationale Altruisten Mannheim e.V. and the University of Luxembourg. We would like to thank C. Akil and D. Mısra Gürol for assistance in instrument adaptation, A. Heske and S. Hörberg for developing the filler task and for helping to implement the pilot study in Qualtrics, as well as A. van Stekelenburg and E. Maibach for useful feedback on an earlier draft of this manuscript. Computational resources were provided by the e-INFRA CZ project (ID:90140, F.B.), supported by the Ministry of Education, Youth and Sports of the Czech Republic. This project received funding from an internal small expenses budget from the Social Psychology Program, Department of Psychology, Faculty of Social and Behavioural Sciences, University of Amsterdam (B.R.); Columbia University's Office for Undergraduate Globe Education (K.R.); and the National Science Foundation, Directorate for Social, Behavioral and Economic Sciences (no. 2218595, K.R.). The funders had no role in study design, data collection and analysis, decision to publish or preparation of the manuscript.

## Author contributions

B.V. and S.J.G. contributed to conceptualization, methodology, analysis planning, investigation, project administration and writing the original draft and revisions. F. Bartoš was involved in software, formal analysis, resources, data curation, visualization and writing of the original draft. M.P.W. undertook supervision and reviewed and edited the manuscript. B.T.R. and F.v.H. were involved in supervision and funding acquisition and reviewed and edited the manuscript. F.S. undertook methodology (translation), investigation (data collection) and project administration and reviewed and edited the manuscript. B.A., A.A., J.B., F. Berglund, A.B.Z., M.B., A. Catania, A. Chen, M.C., E.F., J.G., B.H.-I., G.J., S.J., J.K., Ž.K., A.K., T.L.A., J.L., D.N., S.N.-O., E.N., C.P., R.P., S.P., F.P., A. Remsö, A. Roh, B.R., J.S., M.S., V.V.N., R.W., K.W., M.W., C.Y.W. and M.Y. contributed to methodology (translation) and investigation (data collection) and reviewed and edited the manuscript. K.R. undertook supervision, funding acquisition, project administration and investigation and reviewed and edited the manuscript. S.v.L. took part in supervision and methodology and reviewed and edited the manuscript.

## Competing interests

The authors declare no competing interests.

## Additional information

**Correspondence and requests for materials** should be addressed to Sandra J. Geiger or Kai Ruggeri.

Bojana Većkalov [1,42], Sandra J. Geiger [2,42] ✉, František Bartoš [1,3], Mathew P. White [4], Bastiaan T. Rutjens [1], Frenk van Harreveld[1,5], Federica Stablum [6], Berkan Akın[1,7], Alaa Aldoh [1], Jinhao Bai [8], Frida Berglund [9,10], Aleša Bratina Zimic [11], Margaret Broyles [12], Andrea Catania [13], Airu Chen [14], Magdalena Chorzępa[15], Eman Farahat [16], Jakob Götz [15,17], Bat Hoter-Ishay [14], Gesine Jordan [18], Siri Joustra [19], Jonas Klingebiel [20], Živa Krajnc [21,22], Antonia Krug[23], Thomas Lind Andersen [24], Johanna Löloff [25], Divya Natarajan[26], Sasha Newman-Oktan [27], Elena Niehoff [28], Celeste Paerels[29], Rachel Papirmeister [30], Steven Peregrina[14], Felicia Pohl [31], Amanda Remsö [32], Abigail Roh [14], Binahayati Rusyidi [33], Justus Schmidt [7], Mariam Shavgulidze [34], Valentina Vellinho Nardin [35], Ruixiang Wang[36], Kelly Warner[14], Miranda Wattier [14], Chloe Y. Wong[37], Mariem Younssi [38], Kai Ruggeri [39,40] ✉ & Sander van der Linden [41]

[1]Department of Psychology, Faculty of Social and Behavioural Sciences, University of Amsterdam, Amsterdam, the Netherlands. [2]Environmental Psychology, Department of Cognition, Emotion and Methods in Psychology, Faculty of Psychology, University of Vienna, Vienna, Austria. [3]Institute of Computer Science of the Czech Academy of Sciences, Prague, Czech Republic. [4]Cognitive Science Hub, University of Vienna, Vienna, Austria. [5]National Institute for Public Health and the Environment (RIVM), Bilthoven, the Netherlands. [6]University of Trento, Department of Psychology and Cognitive Science, Trento, Italy. [7]University of Mannheim, Department of Psychology, School of Social Sciences, Mannheim, Germany. [8]Liberal Arts Program, Faculty of Humanities, Tel Aviv University, Tel Aviv, Israel. [9]Department of Psychology, Uppsala University, Uppsala, Sweden. [10]Department of Women's and Children's Health, Uppsala University, Uppsala, Sweden. [11]Department of Medicine and Psychology, Sapienza University of Rome, Rome, Italy. [12]Department of Industrial Engineering and Operations Research, Fu Foundation School of Engineering and Applied Science, Columbia University, New York, NY, USA. [13]Department of Psychology, University of Malta, Msida, Malta. [14]Department of Psychology, Columbia University, New York, NY, USA. [15]Department of Psychology, Faculty of Behavioural and Movement Sciences, Vrije Universiteit Amsterdam, Amsterdam, the Netherlands. [16]Department of Psychology, Behavioural and Economic Science, University of Warwick, Coventry, UK. [17]Motivation Psychology, Department of Occupational, Economic and Social Psychology, Faculty of Psychology, University of Vienna, Vienna, Austria. [18]Department of Behavioural and Cognitive Sciences, University of Luxembourg, Esch-sur-Alzette, Luxembourg. [19]Department of Psychology, Faculty of Social Sciences, Radboud University, Nijmegen, the Netherlands. [20]School of General Studies, Columbia University, New York, NY, USA. [21]Department of Psychology, University of Ljubljana, Ljubljana, Slovenia. [22]Department of Psychology, University of Maribor, Maribor, Slovenia. [23]Institute of Psychology, University of Innsbruck, Innsbruck, Austria. [24]Child and Adolescent Mental Health Center, Copenhagen University Hospital—Mental Health Services CPH, Copenhagen, Denmark. [25]Department of Psychology, Heidelberg University, Heidelberg, Germany. [26]Department of Cognitive Science, Barnard College, Columbia University, New York, NY, USA. [27]Program in Cognitive Science, Columbia University, New York, NY, USA. [28]Environmental Policy Group, Wageningen University & Research, Wageningen, the Netherlands. [29]Department of Ecology, Evolution and Environmental Biology, Columbia University, New York, NY, USA. [30]Department of Cognitive Science, Columbia University, New York, NY, USA. [31]Faculty of Psychology, Warsaw International Studies in Psychology, University of Warsaw, Warsaw, Poland. [32]Department of Psychology, Faculty of Education, Kristianstad University, Kristianstad, Sweden. [33]Social Welfare Department & Center for CSR, Social Entrepreneurship & Community Empowerment, FISIP, Universitas Padjadjaran, Jatinangor-Sumedang, Indonesia. [34]Institute of Psychology, Eötvös Loránd University, Budapest, Hungary. [35]Department of Psychology, Faculty of Psychology and Education Sciences, University of Porto, Porto, Portugal. [36]Columbia College, Columbia University, New York, NY, USA. [37]Barnard College, Columbia University, New York, NY, USA. [38]LAPCOS, Université Côte d'Azur, Nice, France. [39]Department of Health Policy and Management, Mailman School of Public Health, Columbia University, New York, NY, USA. [40]Policy Research Group, Centre for Business Research, Judge Business School, University of Cambridge, Cambridge, UK. [41]Department of Psychology, School of the Biological Sciences, University of Cambridge, Cambridge, UK. [42]These authors contributed equally: Bojana Većkalov, Sandra J. Geiger. ✉e-mail: sandra.geiger@univie.ac.at; dar56@cam.ac.uk

# Reporting Summary

## Statistics

For all statistical analyses, confirm that the following items are present in the figure legend, table legend, main text, or Methods section.

| n/a | Confirmed | |
|---|---|---|
| ☐ | ☒ | The exact sample size (*n*) for each experimental group/condition, given as a discrete number and unit of measurement |
| ☐ | ☒ | A statement on whether measurements were taken from distinct samples or whether the same sample was measured repeatedly |
| ☐ | ☒ | The statistical test(s) used AND whether they are one- or two-sided *Only common tests should be described solely by name; describe more complex techniques in the Methods section.* |
| ☐ | ☒ | A description of all covariates tested |
| ☐ | ☒ | A description of any assumptions or corrections, such as tests of normality and adjustment for multiple comparisons |
| ☐ | ☒ | A full description of the statistical parameters including central tendency (e.g. means) or other basic estimates (e.g. regression coefficient) AND variation (e.g. standard deviation) or associated estimates of uncertainty (e.g. confidence intervals) |
| ☐ | ☒ | For null hypothesis testing, the test statistic (e.g. *F*, *t*, *r*) with confidence intervals, effect sizes, degrees of freedom and *P* value noted *Give P values as exact values whenever suitable.* |
| ☐ | ☒ | For Bayesian analysis, information on the choice of priors and Markov chain Monte Carlo settings |
| ☐ | ☒ | For hierarchical and complex designs, identification of the appropriate level for tests and full reporting of outcomes |
| ☐ | ☒ | Estimates of effect sizes (e.g. Cohen's *d*, Pearson's *r*), indicating how they were calculated |

*Our web collection on statistics for biologists contains articles on many of the points above.*

## Software and code

Policy information about availability of computer code

| Data collection | Data collection was conducted using the Qualtrics XM web service platform (version: July/August 2023). |
|---|---|
| Data analysis | The R code for the main analyse and the Bayesian Design analyses are publicly available on the Open Science Framework (https://osf.io/z6quh/). To run the mixed-effects linear regression models, we used the R package BayesFactor. For the mixed-effects cumulative probit regression models, we used Stan and the Rstan R package, and computed the marginal likelihood via bridge sampling using the bridgesampling R package. |

For manuscripts utilizing custom algorithms or software that are central to the research but not yet described in published literature, software must be made available to editors and reviewers. We strongly encourage code deposition in a community repository (e.g. GitHub). See the Nature Portfolio guidelines for submitting code & software for further information.

## Data

Policy information about <u>availability of data</u>

All manuscripts must include a <u>data availability statement</u>. This statement should provide the following information, where applicable:

- Accession codes, unique identifiers, or web links for publicly available datasets
- A description of any restrictions on data availability
- For clinical datasets or third party data, please ensure that the statement adheres to our <u>policy</u>

> The raw and cleaned datasets for all analyses reported in this manuscript are publicly available under a CC-By Attribution International 4.0 license on the Open Science Framework (https://osf.io/z6quh/).

## Research involving human participants, their data, or biological material

Policy information about studies with <u>human participants or human data</u>. See also policy information about <u>sex, gender (identity/presentation), and sexual orientation</u> and <u>race, ethnicity and racism</u>.

| | |
|---|---|
| Reporting on sex and gender | Gender was considered as demographic information and as a covariate in the current study design. Gender was determined based on self-reporting ("What is your gender?", including male, female, other, and prefer not to say as response options). The source data include disaggregated gender data, and consent has been obtained for sharing these individual-level data. Overall, our sample included 57% participants that identified as female, 42% as male, and 1% as other. |
| Reporting on race, ethnicity, or other socially relevant groupings | We assessed ethnicity in our survey to describe our samples. Ethnicity was determined based on self-reporting ("Please choose which best describes you."), with response categories carefully adapted to the country-specific context. Due to the plurality of the overall response options, which makes it impossible to provide a single summary statistic, we did not use this item to describe the samples. |
| Population characteristics | Participants were 57% female, with a mean age of 33.7 (SD = 13.3). Almost 100% of participants had completed some formal education, with 68.1% completing a university degree and 32.7% were current students. Most participants (81.6%) lived in urban areas. |
| Recruitment | We use what we refer to as the Demic-Veckalov (named for Emir Demic and Bojana Veckalov) method for sampling: All collaborators used a range of circulation points, including email lists, discussion boards, and social media pages to recruit as random a sample as possible. This meant we primarily did not use individual pages to recruit, but instead, found recent posts with high engagement (often related to popular media topics except climate change/sustainability) as well as common interest platforms (e.g., Reddit channels). We also contacted universities and other organizations to assist with circulation. The primary forms of bias that this could create would be over-representation of individuals with computers/social media accounts, younger and more educated participants (due to the types of news stories often used as a conduit for recruiting), and individuals that speak the primary local language. |
| Ethics oversight | We obtained ethical approval from the Institutional Review Board from the University of Amsterdam (the Netherlands; protocol FMG-1123) and the University of Porto (Portugal; protocol 2023/06-12). |

Note that full information on the approval of the study protocol must also be provided in the manuscript.

# Field-specific reporting

Please select the one below that is the best fit for your research. If you are not sure, read the appropriate sections before making your selection.

☐ Life sciences   ☒ Behavioural & social sciences   ☐ Ecological, evolutionary & environmental sciences

For a reference copy of the document with all sections, see <u>nature.com/documents/nr-reporting-summary-flat.pdf</u>

# Behavioural & social sciences study design

All studies must disclose on these points even when the disclosure is negative.

| | |
|---|---|
| Study description | A 27-country study testing the effectiveness of scientific consensus messaging on climate change for climate change beliefs, worry, and support for public action. All participants completed an approximately 5-min survey which randomly assigned participants to one of three messages: a control message (97% of dentists recommend brushing your teeth twice per day), the classic scientific consensus message (97% of climate scientists agree that human-caused climate change is happening), and an updated scientific consensus message that additionally presented the agreement among climate scientists that climate change is a crisis (In addition, 88% of climate scientists agree that climate change constitutes a crisis). |
| Research sample | Convenience sample of adults (18 years and older) from 27 countries (57% female; mean age = 33). Samples were not recruited in a way that ensured representativeness, but instead we focused on obtaining a large enough sample from a diverse set of countries that provides sufficiently powered estimates for comparisons across and partly between countries. |

| | |
|---|---|
| Sampling strategy | We used the Demić–Većkalov method (Ruggeri et al., 2022) to obtain convenience samples, as described above under "Recruitment". The overall and per-country sample size was determined based on Bayes Factor Design Analyses (BFDA). Details on the BFDA can be found in the Supplementary Materials 3.5.<br><br>Reference: Ruggeri, K., Panin, A., Vdovic, M., Većkalov, B., Abdul-Salaam, N., Achterberg, J., ... & Toscano, F. (2022). The globalizability of temporal discounting. Nature Human Behaviour, 6(10), 1386-1397. |
| Data collection | All participants completed the study via Qualtrics; no researcher was present at the time of data collection and there were no conditions for blinding. Participants completed the survey in the primary local langauge (in some cases, an additional English version of the survey was offered). |
| Timing | All data were collected between late July 27 and August 4, 2023. |
| Data exclusions | At the participant level, we excluded respondents who failed the attention check (n = 1,032) or finished the survey in less than 2 minutes (n = 0; based on pilot data in which none of the respondents completed a very similar survey in less than 2 minutes). In addition, for paid samples, responses identified as potential bots (Qualtrics variable: Q_RecaptchaScore < .50; n = 3), duplicates (Qualtrics variable: Q_RelevantIDDuplicateScore ≥ 75; n = 9), or fraudulent (Qualtrics variable: Q_RelevantIDFraudScore ≥ 30, as per Qualtrics recommendations; n = 6) were excluded. Given that these are not equally functional and thus informative across countries, we kept respondents for whom these metrics were not recorded. All exclusion criteria were preregistered.<br><br>Participants with missing data were excluded on an analysis-by-analysis basis. Prior to data collection, we expected few missing values due to the forced-response format of all items. This was confirmed, with 130 missing values on belief in the human causation of climate change and 128 missing values on gender.<br><br>We did not define or remove any outliers, as all measures are bounded, which effectively guards us from any outliers. |
| Non-participation | Out of 21,463 people who clicked on the link, 11,702 participants completed the survey. Out of these, 10,527 participants remained after further exclusions (see Data exclusions above). |
| Randomization | Participants were randomly assigned to one of three groups (control, classic scientific consensus, and updated scientific consensus condition). This randomization was implemented via Qualtrics, such that each message was equally often presented across all participants. |

# Reporting for specific materials, systems and methods

We require information from authors about some types of materials, experimental systems and methods used in many studies. Here, indicate whether each material, system or method listed is relevant to your study. If you are not sure if a list item applies to your research, read the appropriate section before selecting a response.

## Materials & experimental systems

| n/a | Involved in the study |
|---|---|
| ☒ | ☐ Antibodies |
| ☒ | ☐ Eukaryotic cell lines |
| ☒ | ☐ Palaeontology and archaeology |
| ☒ | ☐ Animals and other organisms |
| ☒ | ☐ Clinical data |
| ☒ | ☐ Dual use research of concern |
| ☒ | ☐ Plants |

## Methods

| n/a | Involved in the study |
|---|---|
| ☒ | ☐ ChIP-seq |
| ☒ | ☐ Flow cytometry |
| ☒ | ☐ MRI-based neuroimaging |

