## [Peer Review File · Nature Human Behaviour]

Peer Review Information

Journal: Nature Human Behaviour

Manuscript Title: A 27-country test of communicating the scientific consensus on climate change

Corresponding author name(s): Sandra J. Geiger, Kai Ruggeri

Reviewer Comments & Decisions:

Decision Letter, initial version:

29th September 2023

Dear Ms Geiger,

Thank you once again for your manuscript, entitled "A 27-country test of communicating the scientific consensus on climate change," and for your patience during the peer review process.

Your manuscript has now been evaluated by 3 reviewers, whose comments are included at the end of this letter. Although the reviewers find your work to be of interest, they also raise some important concerns. We are interested in the possibility of publishing your study in Nature Human Behaviour, but would like to consider your response to these concerns in the form of a revised manuscript before we make a decision on publication.

To guide the scope of the revisions, the editors discuss the referee reports in detail within the team, including with the chief editor, with a view to (1) identifying key priorities that should be addressed in revision and (2) overruling referee requests that are deemed beyond the scope of the current study. We hope that you will find the prioritised set of referee points to be useful when revising your study. Please do not hesitate to get in touch if you would like to discuss these issues further.

You will see from their comments that all three reviewers raise substantial concerns about the use of convenience sampling in this case and the lack of representativeness in your obtained samples. However, they are divided in their views on the impacts of this weakness and their recommendations regarding publication. Editorially, we agree that this is a serious concern that requires substantial and thorough discussion as a limitation (including in the abstract, not just the main text). However, in light of the divided views among the reviewers, we believe that there is a path forward for your paper provided that you thoroughly and transparently discuss the implications of this issue for your results.

In addition to revising to address this key point, we also ask that you address the following (as well as all other reviewer comments):

1) Address reviewer concerns regarding the small effect sizes found, ensuring that effect sizes are transparently discussed and that interpretations and conclusions reflect the magnitude of the effects.

2) Include additional discussion and carry out additional analyses (where feasible given sample sizes) to address reviewer concerns about the need for further cross-cultural insight.

3) Address Reviewer 1's concerns regarding the consistency of your analysis with the pre-analysis plan, ensuring that details are clear and any deviations are transparent and justified. All preregistered analyses must be reported, except if shown to be unfeasible or scientifically unsound.

In sum, we invite you to revise your manuscript taking into account all reviewer and editor comments. We are committed to providing a fair and constructive peer-review process. Do not hesitate to contact us if there are specific requests from the reviewers that you believe are technically impossible or unlikely to yield a meaningful outcome.

We hope to receive your revised manuscript within two months. I would be grateful if you could contact us as soon as possible if you foresee difficulties with meeting this target resubmission date.

- Include a "Response to the editors and reviewers" document detailing, point-by-point, how you addressed each editor and referee comment. If no action was taken to address a point, you must provide a compelling argument. When formatting this document, please respond to each reviewer comment individually, including the full text of the reviewer comment verbatim followed by your response to the individual point. This response will be used by the editors to evaluate your revision and sent back to the reviewers along with the revised manuscript.
- Highlight all changes made to your manuscript or provide us with a version that tracks changes.

[REDACTED]

We look forward to seeing the revised manuscript and thank you for the opportunity to review your work. Please do not hesitate to contact me if you have any questions or would like to discuss these revisions further.

Sincerely,

[REDACTED]

Reviewer expertise:

Reviewer #1: cross-cultural psychology, climate beliefs

Reviewer #2: applied statistics

Reviewer #3: communication, climate change messaging

REVIEWER COMMENTS:

Reviewer #1:

Remarks to the Author:

Broadening our cross-cultural understanding of the impact of framing climate change is an important research question, and pre-registration of research is highly desirable, so on these counts the author(s) work should be commended. The results also provide some information useful to a broad audience. Despite this, I have some concerns about the research that I describe below, although I think it's possible to address them if a revision is offered.

I first note that I have little experience with Bayesian analyses beyond the basics so I cannot critique this aspect of the manuscript – I appreciate there is an introduction in the Supplementary Materials but as I already understood the (very) basic elements of the approach, and the more advanced descriptions related to specific model choices, I believe that the detailed analytical choices require more specialist knowledge/evaluation and hope that another reviewer can provide this expertise.

One concern is that the measures serving as the main outcomes appeared more like manipulation/comprehension checks, e.g., after being told the scientific consensus was 97% they were asked what the scientific consensus was. While the authors did note that backlash effects might be possible, in the context of the survey it's likely to be seen by participants as "did you read what you were just told?". To me, the other outcome variables (e.g., own belief, whether climate change is a crisis, support for action), as well as moderation effects, are more theoretically and practically interesting for the field. It would be ideal if the authors were able to address this concern, even if just by discussing it as a potential limitation.

Related to this, the conclusion in the abstract (and throughout) that consensus messaging is effective seems too optimistic – it could be argued that it is not effective if it makes no difference to support for public action, and if other (non-manipulation-check-adjacent) effect sizes are very small, for instance

communicating scientific consensus about a crisis does not increase average beliefs that climate change is a crisis. A more nuanced conclusion about effectiveness in the abstract/discussion seems appropriate, including greater attention to effect sizes.

In the introduction (lines 46-48) two papers are cited for a claim that consensus messaging was found to : "...increase several pro-climate attitudes (Hedge's $g = 0.08 - 0.12$), namely personal beliefs in climate change, worry, and support for public climate action." Of the 2 papers, I think only one examined policy support (the Rode et al., paper) and the effect size was not significantly different from zero. It would be good if these findings could be reported with a bit more detail/accuracy (which would also make correspondence with the present findings easier to evaluate).

I appreciate how difficult it can be to obtain cross-cultural samples, especially when relying on local collaborators rather than (rare) massive grants to fund data collection, so I think some leeway on sample representativeness/size is appropriate. At the same time, people who react negatively to the materials (e.g., consensus messaging because they react negatively to climate change) may be more likely to exit the survey in these contexts – the "backlash" is abandoning the survey rather than not reporting updated beliefs. To address this concern, it should be possible in Qualtrics to identify how many dropped out (establishing a completion rate) and possibly even how many dropped out at the point of seeing the manipulations. It might even be possible to identify whether those who dropped out were more likely to have lower pre-intervention consensus/agreement scores. This type of information would help rule out concerns that a lack of backlash (and more generally positive effects) were influenced by those more open to the materials/topic being more likely to complete the whole survey.

Given that cross-cultural relevance was at the center of the claimed contribution of this work, it was disappointing that the introduction barely considered country/culture beyond generalizability (and compare the extent of consideration for Q6 in Table 1 compared to other research questions). Even at a very basic level there are theories/ideas about why culture might matter, such as individualism-collectivism. For example, if you consider consensus as a type of norm, there is evidence indicating that norms predict action more strongly for those who are more collectivistic (<https://doi.org/10.1177/014616722110072>). More nuanced consideration might even consider differences between vertical and horizontal Ind-Col (Triandis (1996). The psychological measurement of cultural syndromes. *American Psychologist*, 51, 407-415), with the views of experts potentially more influential in vertical cultures where authority/hierarchy is more important. By raising these possible cultural factors I am not saying these specific ideas/findings should be used, but they are just examples to illustrate possible solutions to the problem that the cultural/country element of the manuscript seems very underdeveloped. In my view the manuscript would benefit from greater engagement with theories of cultural differences and relevant cross-cultural research on climate change issues (which is substantial).

This minimal engagement with culture may have contributed to the lack of consideration of cultural differences in the results even when they were identified. With 27 countries it would be possible to perform meta-regression or incorporate level 2 predictors in the mixed models to identify if country-level measures (e.g., Ind-Col) help explain country-level variation in effects (for a meta-regression example, see <https://doi.org/10.1038/s41893-022-00902-y>). This could even be more informative than the (lack of) per-country recommendations as it based on more generalizable country differences. Moreover, it would be a useful service to future research to report in Supplementary Materials not just the meta-analytic effect size for heterogeneity, but also effect sizes for each country

so readers can get a better sense of country differences.

I found the labels for the two questions (classic/updated in Fig 2) quite vague. Alternatives might better highlight the content of the difference, e.g., reality and reality+crisis.

I understand the desire for brevity when reporting/interpreting results, but in this case I found the interpretation of means a little misleading. For example, in lines 182-183 it is stated that: "Across all 27 countries (N = 10,527), people consistently underestimate the scientific consensus". To me that is an inaccurate interpretation of means, as an average does not indicate consistency within a sample. To illustrate, an average of 80% could be found by 80% of people reporting 100% consensus and 20% reporting 0% - so only a minority (20%) underestimated. What is reported is not consistency of underestimating (that would be established by showing that the proportion of each sample that underestimated was 100% - if that is the desired claim then the proportion per sample should also be reported), but that the average consensus reported in each sample was below the comparative level of scientific consensus. I think this can be addressed by slight rephrasing of interpretations, and ideally (in my view) by providing the proportion of each sample who underestimated as I think this is useful and meaningful information.

For the sections starting on line 205 "Effectiveness of the classic scientific consensus message" and line 249: "Effectiveness of the updated scientific consensus message", it was not clear from the writing whether this is comparing just post-intervention measures or pre-post change. I actually expected that this section would be about pre-post comparisons at this seems most relevant to assessing "effectiveness", and controlling for pre-treatment measurement in all models was specified in the preregistration. But based on later sections it seemed more likely this was post-intervention only (even though the Supplementary Materials mentioned that all models will control for pre-treatment measures (SM lines 271-272). This could be easily clarified by a short statement (or perhaps just pointing to the location of this information in the article and ideally making it more prominent).

But if I am correct that the analyses are post-intervention only, to me the pre-post change is the most appropriate analysis for assessing message effectiveness (as seemed to be the case in the pre-registration/Supplementary Materials), and thus deserves much more attention than minor coverage in the exploratory analysis section.

Reference 11 is the same as reference 6, and 65/67 are also duplicated. There may be others and this should be checked more closely.

Reviewer #2:
Remarks to the Author:

A 27-country test of communicating the scientific consensus on climate change

How to successfully convince the general public about the main cause and severity of climate change is one of the main societal challenges we're facing. The current manuscript provides a systematic, pre-registered approach to better understand this challenge.

The set up of the study is sound and explained clearly and in sufficient detail for replication purposes. The statistical analyses are appropriate for the design and research question and the results are interpreted clearly. A particularly strong point of the study is that the authors do not focus solely on Western countries: as climate change is a global challenge, it requires a global approach.

I do have one major concern, and this could be a 'deal breaker'. My major concern deals with how the data were collected: this has largely been done through convenience sampling. Even the selection of participating countries is some kind of convenience sample from the 'population' of all countries in the world. This raises serious concerns w.r.t. representativeness of the sample. Especially since one of the main goals is checking whether the proportion of the general population that agrees that climate change is anthropogenic aligns with the scientific consensus percentage of 97%, representivity of the sample is a must.

By recruiting participants via snowballing, social media and mailing lists the risk of non-representativeness is large. The demographics in Table 2 also contain a number of indications that the sample is far from representative. The participants largely seem to be young (mean age 33.7) and highly educated (68% has university education). Furthermore in 14 out of 27 countries, the male/female ratio is so unbalanced that either men or women constitute at least 60% of the sample. If these demographics are biased, what's to say that the percentages as presented in e.g. Figure 1 aren't severely biased as well? That you adjust for gender and other demographics in the analysis is helpful but mitigates this problem only partially.

Even though the rest of the work seems outstanding and state of the art, the adagium 'garbage in, garbage out' hold true here. Without guarantees on the quality of the data, the quality of the results cannot be guaranteed...

As said, the remainder of the manuscript is of excellent quality. I have a number of minor issues and questions for clarification:

- Fig. 1: the countries are now sorted alphabetically. It is more insightful to sort them on basis of their score (thus either the red or blue percentage).
- The differences between countries could be explored more. Fig.1 gives results per country, but this is not interpreted in detail for (systematic) differences. Do, for instance, the ~10 European countries yield systematically lower/higher results than e.g. the Asian or African countries? From all countries with a 'sufficient' sample size, China clearly scores lower than the rest; how could this be explained? Etc.

Reviewer #3:

Remarks to the Author:

The authors are to be congratulated on a superbly written paper that is likely to be of wide interest within the climate change communication community. The literature review of the paper could serve as a model for graduate students to follow when they begin publishing. And the discussion was thoughtful and will be useful for strategists planning climate information campaigns.

My single concern about the study is its use of convenience samples, and the ways in which this may have affected the results. There were no quotas set to complete the survey, which would have improved the representativeness of the samples. Comparisons between Table 2 in the paper, which

shows the demographics of each country's sample, and Table 3.1 in the supplementary materials, which shows population statistics for each country, reveal large discrepancies between the samples' and the nations' demographics. For example, in 25 of the 27 countries, more than half of the sample has a university degree, while the national data show that only 5 of the 27 countries have populations in which half have completed post-secondary education. In the Egyptian sample, 96% have a university degree, while nationally, only 13% have post-secondary degrees. Discrepancies are apparent in gender and urbanicity as well: Three-quarters of the samples in Georgia, Lebanon & Serbia are female; 92% of the Egyptian sample lives in urban areas, compared to 43% of the population.

Quotas would also have been helpful for political ideology: As right-leaning people have been shown to be more responsive to consensus messages, samples which do not reflect the national balance will not assess the effects of the messages accurately.

The demographic differences between the samples and nations may have affected the results in multiple ways. For example, if people with more education have more familiarity with the scientific consensus, the mean pre-intervention consensus perceptions shown in Figure 1 are overestimated for the nations with large sample-population discrepancies. And the reported effects of the messages are being underestimated, as the people who are unfamiliar with the messages are underrepresented. I believe discussion of this limitation of the research should be added to the paper.

Despite this weakness, this is an excellent study overall that definitely merits publication.

Smaller points:

- I believe the data from Lebanon should be deleted from the paper, as the sample is far too small to yield meaningful results ($n=9$); Tunisia's data is also questionable due to the small sample ($n=92$).
- I couldn't locate the OSF file cited in line 178; please add the file name to help readers find the data.
- On lines 434-435, the authors state that the consensus message does not extend to support for public action. This should be amended to say the message does not extend DIRECTLY to policy support. The Gateway Belief Model does not posit a direct effect, which – as the authors note – their data cannot test.
- While overall, the writing is excellent, there were several sentences I found confusing:
 - o Footnote 1
 - o The sentence in lines 133-135
 - o The sentence in lines 389-391
 - o The sentence in lines 467-469

Author Rebuttal to Initial comments

REVIEWER COMMENTS:

Reviewer #1:

Remarks to the Author:

Broadening our cross-cultural understanding of the impact of framing climate change is an important

research question, and pre-registration of research is highly desirable, so on these counts the author(s) work should be commended. The results also provide some information useful to a broad audience. Despite this, I have some concerns about the research that I describe below, although I think it's possible to address them if a revision is offered.

I first note that I have little experience with Bayesian analyses beyond the basics so I cannot critique this aspect of the manuscript – I appreciate there is an introduction in the Supplementary Materials but as I already understood the (very) basic elements of the approach, and the more advanced descriptions related to specific model choices, I believe that the detailed analytical choices require more specialist knowledge/evaluation and hope that another reviewer can provide this expertise.

We very much appreciate the positive feedback and summary of our work. We also note the attention to extremely important details within each major aspect and hope that by providing detailed responses and revisions to the text, you feel these have been appropriately described, resolved, and otherwise addressed.

1. One concern is that the measures serving as the main outcomes appeared more like manipulation/comprehension checks, e.g., after being told the scientific consensus was 97% they were asked what the scientific consensus was. While the authors did note that backlash effects might be possible, in the context of the survey it's likely to be seen by participants as “did you read what you were just told?”. To me, the other outcome variables (e.g., own belief, whether climate change is a crisis, support for action), as well as moderation effects, are more theoretically and practically interesting for the field. It would be ideal if the authors were able to address this concern, even if just by discussing it as a potential limitation.

We agree that the more consequential goal of the consensus messaging interventions is to change personal climate change beliefs, and possibly support for public action regarding climate change. However, perceived scientific consensus is an important construct in its own right, because it is the target of diverging theoretical predictions in scientific consensus research (Bertoldo et al., 2019; van der Linden 2021) and therefore more than a manipulation/comprehension check. On the one hand, research based on the Gateway Belief Model postulates that consensus perception is a “gateway” to altering downstream (i.e., the above-mentioned more ultimate) climate change beliefs, worry, and support for public action (van der Linden, 2021). For this to happen, it is necessary that participants update their perceptions of the scientific consensus. In that vein, post-treatment consensus perceptions are a crucial part of the so-called “estimate-and-reveal” technique, in which asking for the consensus estimates before and after the messaging intervention serves to highlight the discrepancy between what the participants thought was the case and what is really the case (Myers et al., 2015). On the other hand, the influential cultural cognition thesis of scientific consensus postulates that consensus perceptions are heavily *shaped by* already-held climate change attitudes and general value orientations (Kahan et al., 2011)--potentially contributing to backfiring (or no) effects of the intervention specifically on consensus perceptions. Indeed, the

expectation here is that identity-protective cognition would lead people to dismiss the scientific consensus when the consensus is not congenial to their worldviews.

Additionally, we would like to emphasise that there was a distractor task between the pre- and post-test measurement to reduce demand effects (noted on p. 22). Furthermore, if people generally understood it as a manipulation check, we would then expect virtually everyone in the final sample (i.e., people who passed an attention check) to have answered “97%”. This, however, is not the case, as 45% in the experimental conditions answered 97%. This clearly indicates that updating occurred, but speaks against seeing this question as a simple comprehension/manipulation check.

Finally, we would like to posit that citizens being correctly informed about the scientific consensus (i.e., correcting misperceptions) on pressing societal issues is a valuable democratic goal in and of itself.

Bertoldo, R., Mays, C., Böhm, G., Poortinga, W., Poumadère, M., Tvinnereim, E., Arnold, A., Steentjes, K., & Pidgeon, N. (2019). Scientific truth or debate: On the link between perceived scientific consensus and belief in anthropogenic climate change. *Public Understanding of Science*, 28(7), 778-796. <https://doi.org/10.1177/0963662519865448>

Kahan, D. M., Jenkins-Smith, H., & Braman, D. (2011). Cultural cognition of scientific consensus. *Journal of Risk Research*, 14(2), 147–174. <https://doi.org/10.1080/13669877.2010.511246>

Myers, T. A., Maibach, E., Peters, E. & Leiserowitz, A. Simple messages help set the record straight about scientific agreement on human-caused climate change: The results of two experiments. *PLOS ONE* 10, e0120985 (2015).

van der Linden, S. (2021). The Gateway Belief Model (GBM): A review and research agenda for communicating the scientific consensus on climate change. *Current Opinion in Psychology*, 42, 7–12. <https://doi.org/10.1016/j.copsyc.2021.01.005>

2. Related to this, the conclusion in the abstract (and throughout) that consensus messaging is effective seems too optimistic – it could be argued that it is not effective if it makes no difference to support for public action, and if other (non-manipulation-check-adjacent) effect sizes are very small, for instance

communicating scientific consensus about a crisis does not increase average beliefs that climate change is a crisis. A more nuanced conclusion about effectiveness in the abstract/discussion seems appropriate, including greater attention to effect sizes.

Thank you for this point, we agree that nuanced effect size considerations are important. We have therefore amended the abstract to explicitly include effect sizes: “the classic message substantially reduces misperceptions ($d = 0.47$) and slightly increases climate change beliefs ($d = 0.05-0.09$) and worry ($d = 0.04$), but not support for public action.” Additionally, in the abstract we now mention specifically for which outcomes scientific consensus messaging is an effective tool: “scientific consensus messaging is an effective, non-polarizing tool for changing misperceptions, beliefs, and worry climate change communication across different audiences.”

Even though we agree that, from a practical application standpoint, it would have been more useful if our results pointed to a direct effect of consensus messaging on support for public action, we disagree that this is the only relevant outcome for several reasons. First, as we mentioned in our reply to your first point, citizens being correctly informed about the scientific consensus (i.e., correcting misperceptions) on pressing societal issues is a valuable democratic goal in and of itself, particularly because changes in perceived societal norms often precede changes in personal attitudes (e.g., on same-sex marriage, see Paluck & Tankard, 2017). Likewise, increasing personal belief in the reality and anthropogenic nature of climate change is arguably a favorable outcome, given that it is aligning people's attitudes with the scientific reality and thus enabling people to base their consequent decisions based on a better-informed attitude. As Braman and Kahan (2003) noted: “individual opinions influence political outcomes through aggregation. Even a modest amount of variation in opinion across individuals will profoundly influence collective deliberations”.

As for the belief in crisis specifically, we now added an exploratory analysis comparing belief in crisis in the classic vs. control group (p.11): *Additionally, we explore whether the classic message influences belief in climate change as a crisis (not preregistered). We find that the classic scientific consensus message increases belief in climate change as a crisis ($BF_{+0} = 35.803$; Cohen's $d = 0.06$, 95% CI [0.01, 0.10]), with evidence against between-country heterogeneity ($BF_{10} = 5.18 \times 10^{-10}$, $\tau_c = 0.00$, 95% CI [0.00, 0.12]).*

We also now explicitly address the weak effect of the updated condition on crisis belief (p. 14): *Notably, we found only weak evidence for a positive effect on personal belief that climate change is a crisis (Cohen's $d = 0.04$). [...] These findings might be due to participants' perceptions of the scientific agreement being already relatively accurate prior to message exposure.*

As for effect sizes on personal beliefs, we agree they are nominally quite small. However, we argue they are nevertheless meaningful and have added a paragraph in the discussion to elaborate (p. 15): *Second, this study finds nominally small effects⁶⁶ of scientific consensus messaging. However, these effects are in line with previous research^{26,27} and can be practically relevant⁶⁷, as the intervention is easily scalable to reach many people. Considering further how brief the message is, and that it does not directly target personal climate change beliefs, these effects can be seen as rather notable⁶⁸. Targeting specific subgroups, such as those who are most likely to underestimate the consensus, might also increase its effectiveness.*

Kahan, D. M., & Braman, D. (2003). Caught in the crossfire: A defense of the cultural theory of gun-risk perceptions. *University of Pennsylvania Law Review*, 151(4), 1395-1416.

Tankard, M. E., & Paluck, E. L. (2017). The Effect of a Supreme Court Decision Regarding Gay Marriage on Social Norms and Personal Attitudes. *Psychological Science*, 28(9), 1334-1344.
<https://doi.org/10.1177/0956797617709594>

3. In the introduction (lines 46-48) two papers are cited for a claim that consensus messaging was found to : "...increase several pro-climate attitudes (Hedge's $g = 0.08 - 0.12$), namely personal beliefs in climate change, worry, and support for public climate action." Of the 2 papers, I think only one examined policy support (the Rode et al., paper) and the effect size was not significantly different from zero. It would be good if these findings could be reported with a bit more detail/accuracy (which would also make correspondence with the present findings easier to evaluate).

This is a very good point, thank you. We now went back to the Rode et al. paper and found that the overall effect of consensus interventions on climate change attitudes (not support for public action specifically) was $g = 0.09$, 95% CI [0.05, 0.13], $p = .004$. Regarding support for public action, this meta-analysis did not find any intervention effects (including scientific consensus messages but also other interventions). We corrected this in the manuscript and also made sure to emphasise which finding refers to which outcome (p. 2): *In addition, two meta-analyses show that informing people about the scientific consensus can substantially reduce consensus misperceptions (Hedge's $g = 0.56$)²⁶ and—to a smaller extent—increase several pro-climate attitudes (Hedge's $g = 0.09 - 0.12$)^{26,27}, namely personal beliefs in climate change and worry. One of these meta-analyses shows that messaging interventions—including but not limited to scientific consensus messages—had no effect on support for public action²⁷.*

4. I appreciate how difficult it can be to obtain cross-cultural samples, especially when relying on local collaborators rather than (rare) massive grants to fund data collection, so I think some leeway on sample representativeness/size is appropriate. At the same time, people who react negatively to the materials (e.g., consensus messaging because they react negatively to climate change) may be more likely to exit the survey in these contexts – the “backlash” is abandoning the survey rather than not reporting updated beliefs. To address this concern, it should be possible in Qualtrics to identify how many dropped out (establishing a completion rate) and possibly even how many dropped out at the point of seeing the manipulations. It might even be possible to identify whether those who dropped out were more likely to have lower pre-intervention consensus/agreement scores. This type of information would help rule out concerns that a lack of backlash (and more generally positive effects) were influenced by those more open to the materials/topic being more likely to complete the whole survey.

Many thanks for acknowledging the difficulties of many-county projects and for your actionable suggestions. We now report the number of people who dropped out in the manuscript (p. 17-18): *Of 21,462 individuals who clicked on the link, 11,702 participants completed the study while 676 were filtered out at the beginning of the survey because they did not reside in any of the 27 countries. Out of all people who dropped out, most did so after seeing the informed consent (2,687; 29.6%), after the introduction that they are randomly assigned to one topic but before seeing that this topic is climate change (804; 8.8%), and right after the control/intervention message (931; 10.2%). Consensus and agreement perceptions of individuals in the two intervention conditions who dropped out directly after seeing the message (consensus: 78.4% and 79.1% in the classic and updated condition; agreement: 75.3% and 77.1%) are slightly lower compared to perceptions of those who completed the study (consensus: 84.2% and 84.5%; agreement: 83.4% and 83.3%)--indicating selective dropout. However, the dropout rate (i.e., number of dropouts directly after seeing the control/intervention message vs. number of overall dropouts) is comparable between both intervention (classic: 267; 2.9%, updated: 296; 3.3%) and control conditions (368; 4.1%), even slightly higher in the control, indicating that the dropout is unlikely to result from a specific backfire of the consensus messages and is suggestive of a more general tendency of less motivated participants to trickle out of the survey in its initial stages.*

Pre-intervention consensus and agreement perceptions seem to be slightly lower for those who dropped out right after seeing the intervention (but also control) messages. Since our results show that scientific consensus messages are generally more effective for people with lower pre-intervention perceptions, we might underestimate the true effect size of the interventions. We now note this in the discussion in the limitations section (p. 15): *On the one hand, our social-media-based sampling approach may have led us to underestimate the intervention effects because, for example, younger and more educated individuals are more likely to believe in climate change⁶⁷.*

While this is something to note, this would most likely not be fully alleviated by using representative samples, at least ones obtained from online panels. This is because if people indeed drop out because of a negative reaction to the survey material, they would be replaced by a different participant with a similar demographic profile, but not necessarily the same predisposition for a negative reaction to the survey material.

5. Given that cross-cultural relevance was at the center of the claimed contribution of this work, it was disappointing that the introduction barely considered country/culture beyond generalizability (and compare the extent of consideration for Q6 in Table 1 compared to other research questions). Even at a very basic level there are theories/ideas about why culture might matter, such as individualism-collectivism. For example, if you consider consensus as a type of norm, there is evidence indicating that norms predict action more strongly for those who are more collectivistic (<https://doi.org/10.1177/014616722110072>). More nuanced consideration might even consider differences between vertical and horizontal Ind-Col (Triandis (1996). The psychological measurement of cultural syndromes. *American Psychologist*, 51, 407-415), with the views of experts potentially more influential in vertical cultures where authority/hierarchy is more important. By raising these possible cultural factors I am not saying these specific ideas/findings should be used, but they are just examples to illustrate possible solutions to the problem that the cultural/country element of the manuscript seems very underdeveloped. In my view the manuscript would benefit from greater engagement with theories of cultural differences and relevant cross-cultural research on climate change issues (which is substantial).

Thank you for this comment and the actionable suggestions on how to incorporate culture considerations. After considering previous research addressing the interplay between culture on the one hand, and norm adherence and message persuasiveness/credibility on the other, we have added the following to the introduction (p. 4): *If the effectiveness of the intervention varies across countries, this could be due to country-level differences in collectivism and power distance³⁹. Consensus information is essentially a descriptive norm, and norms have been shown to be more predictive of support for environmental policy in collectivistic compared to individualistic cultures⁵⁰. Considering further that scientific consensus messages are an expert norm, it is possible that they are more effective in cultures with higher power distance, where greater weight is given to source expertise/authorities⁵¹⁻⁵³. However, more precise predictions are difficult due to the lack of studies on scientific consensus interventions outside the US^{12,30-33}.*

Please see our reply to the next point for information on how we addressed this from an empirical/analysis standpoint.

6. This minimal engagement with culture may have contributed to the lack of consideration of cultural differences in the results even when they were identified. With 27 countries it would be possible to

perform meta-regression or incorporate level 2 predictors in the mixed models to identify if country-level measures (e.g., Ind-Col) help explain country-level variation in effects (for a meta-regression example, see <https://doi.org/10.1038/s41893-022-00902-y>). This could even be more informative than the (lack of) per-country recommendations as it based on more generalizable country differences.

Many thanks for this suggestion. We now test whether the interventions' effectiveness on consensus/agreement perceptions is moderated by two of Hofstede's dimensions (Hofstede, 2011), (a) collectivism-individualism and (b) power distance. We only test the moderation of the intervention effects on consensus/agreement perceptions because there is no substantial cross-country heterogeneity in the other effects (beliefs, worry, and support for public action). We would also like to note that potential moderation effects would need to be fairly large to be detected, given that our sample includes 27 countries.

For the classic consensus message, we find moderate and strong evidence against any moderation of the effect on consensus perceptions by individualism-collectivism ($BF_{10} = 0.20$; $b = -0.05$, 95% CI [-0.11, 0.01]) and power distance ($BF_{10} = 0.08$; $b = 0.03$, 95% CI [-0.03, 0.10]) respectively. This was also the case for the updated consensus message: we find moderate evidence against a moderation by individualism-collectivism ($BF_{10} = 0.18$, $b = -0.04$, 95% CI [-0.10, 0.02]) and power distance ($BF_{10} = 0.16$, $b = 0.05$, 95% CI [-0.01, 0.11]) on consensus perceptions. For agreement perceptions changes in the updated (vs. control) condition, the results are inconclusive, with weak evidence against a moderation by individualism-collectivism ($BF_{10} = 0.35$, $b = -0.07$, 95% CI [-0.15, 0.02]) and weak evidence for a moderation by power distance ($BF_{10} = 1.53$, $b = 0.08$, 95% CI [0.02, 0.15]).

We report these analyses and results in the supplementary materials (Supplemental Results 2.6)

Hofstede, G. (2011). Dimensionalizing Cultures: The Hofstede Model in Context. *Online Readings in Psychology and Culture*, 2(1). <https://doi.org/10.9707/2307-0919.1014>

7. Moreover, it would be a useful service to future research to report in Supplementary Materials not just the meta-analytic effect size for heterogeneity, but also effect sizes for each country so readers can get a better sense of country differences.

We have included effect sizes for each outcome and each country on OSF under outcome-summaries. We now clarify this on p.7: *For future meta-analyses, we provide a summary table of the results for each outcome per country on OSF (<https://bit.ly/44hKe7F>; subfolder outcome-summaries in the Data & Analyses tab).*

To clarify, there is a separate .csv file for each outcome to simplify readability, as we provide detailed descriptive statistics in addition to Cohen's *ds* in these files.

8. I found the labels for the two questions (classic/updated in Fig 2) quite vague. Alternatives might better highlight the content of the difference, e.g., reality and reality+crisis.

Thank you for this suggestion. We did consider changing it, however we decided against it due to the reality+crisis being potentially confusing in writing. We did however add a note to Fig 2 (p.10 & 11): “‘Classic’ refers to the message communicating the scientific consensus on the reality of climate change. ‘Updated’ refers to the message communicating the scientific consensus on the reality of climate change and the scientific agreement on climate change as a crisis.”

9. I understand the desire for brevity when reporting/interpreting results, but in this case I found the interpretation of means a little misleading. For example, in lines 182-183 it is stated that: “Across all 27 countries (N = 10,527), people consistently underestimate the scientific consensus”. To me that is an inaccurate interpretation of means, as an average does not indicate consistency within a sample. To illustrate, an average of 80% could be found by 80% of people reporting 100% consensus and 20% reporting 0% - so only a minority (20%) underestimated. What is reported is not consistency of underestimating (that would be established by showing that the proportion of each sample that underestimated was 100% - if that is the desired claim then the proportion per sample should also be reported), but that the average consensus reported in each sample was below the comparative level of scientific consensus. I think this can be addressed by slight rephrasing of interpretations, and ideally (in my view) by providing the proportion of each sample who underestimated as I think this is useful and meaningful information.

We agree that the previous phrasing was imprecise. We therefore changed it to reflect that we are discussing sample averages and added information on the percentage of people who underestimated both the consensus and the agreement per country (p. 7-8): *Across all 27 countries (N = 10,527), the scientific consensus that human-caused climate change is happening (97%) is underestimated by, on average, -7.52% (95% CI [-7.79, -7.26]). This underestimation ranges from -15.39% (95% [-17.11, -13.68]) in the Chinese sample to -4.10% (95% [-4.85, -3.35]) in the German sample (Fig. 1). Importantly, 72.2% (95% CI [71.3, 73.0]) of participants underestimate this consensus, ranging from 57.5% (95% CI [52.2, 62.6]) in the US sample to 83.7% (95% CI [79.9, 87.0]) in the Chinese sample (Supplementary Results 2.2).*

Similarly, the scientific agreement that climate change constitutes a crisis (88%) is slightly underestimated by, on average, -1.30% (95% [-1.58, -1.03]). This ranges from underestimation of -8.44% (95% [-21.98, 5.09]) in the Lebanese sample to overestimation of 2.08% (95% [-0.26, 4.41]) in the Tunisian sample, though both samples are rather small. For the sufficiently large samples, misperceptions range from -8.12% (95% CI [-9.85, -6.40]) in the Chinese sample to 2.03% (95% CI [1.28, 2.77]) in the German sample (Fig. 1). However, this scientific agreement is not consistently underestimated. A total of 44.5% participants across all countries (95% CI [43.6, 45.5]) underestimate the crisis agreement, ranging from 29.2% (95% [25.7, 32.9]) in the German sample to 69.5% [65.0, 73.7] in the Chinese sample (Supplementary Results 2.2).

10. For the sections starting on line 205 “Effectiveness of the classic scientific consensus message” and line 249: “Effectiveness of the updated scientific consensus message”, it was not clear from the writing whether this is comparing just post-intervention measures or pre-post change. I actually expected that this section would be about pre-post comparisons at this seems most relevant to assessing “effectiveness”, and controlling for pre-treatment measurement in all models was specified in the preregistration. But based on later sections it seemed more likely this was post-intervention only (even though the Supplementary Materials mentioned that all models will control for pre-treatment measures (SM lines 271-272). This could be easily clarified by a short statement (or perhaps just pointing to the location of this information in the article and ideally making it more prominent).

The reviewer is correct that we measured all personal beliefs, worry and support for public action only post-treatment. In the supplementary materials, as well as the preregistration, we specified that we will control for pre-treatment measurements only for the continuous outcomes (meaning reality and crisis agreement perceptions). We fully followed the preregistered analysis plan. We agree this might have not been obvious from the results section, so we now amended it to emphasize that all analyses are based on post-treatment outcomes (p. 7: *All analyses are based on group differences in post-intervention outcomes.*) and in which analyses we controlled for pre-treatment perceptions (p. 10, 11 & 12).

p. 11: *Controlling for pre-intervention perceptions of the reality consensus, we find extremely strong support for $H1_a$ that post-intervention perceptions of the reality consensus are higher and thus more accurate in the classic scientific consensus compared to the control condition ($BF_{+0} = 2.01 \times 10^{12}$; Fig. 2).*

p. 12: *Controlling for pre-intervention perceptions of the reality consensus and crisis agreement respectively, we find extremely strong support for $H2_a$ and $H2_b$ that perceptions of both the reality consensus ($BF_{+0} = 2.12 \times 10^{12}$; Cohen's $d = 0.45$; 95% CI [0.41, 0.50]; Fig. 2) and the crisis agreement ($BF_{+0} = 1.54 \times 10^5$; Cohen's $d = 0.23$; 95% CI [0.18, 0.28]; Fig. 2) are higher in the updated compared to the control condition, with substantial evidence for relatively small between-country heterogeneity (reality consensus: $BF_{10} = 1,192.27$, $\tau_c = 0.05$, 95% CI [0.00, 0.14]; crisis agreement: $BF_{10} = 1.53 \times 10^8$, $\tau_c = 0.15$, 95% CI [0.09, 0.23]).*

11. 12. But if I am correct that the analyses are post-intervention only, to me the pre-post change is the most appropriate analysis for assessing message effectiveness (as seemed to be the case in the pre-registration/Supplementary Materials), and thus deserves much more attention than minor coverage in the exploratory analysis section.

We now clarified in the manuscript that we focus on post-intervention outcomes (p. 7): *All analyses are based on group differences in post-intervention outcomes.*

We agree that pre-post scores can be informative for measuring messaging effects. However, we employed a between-participants design in careful consideration of the research questions. Although the pre-post change score and ANCOVA analyses can both be used in pre-post test analyses, their focus is slightly different. Change scores focus on the difference between the groups of the *change* itself, while the ANCOVA approach emphasises *treatment* effects, after accounting for where participants are starting from. As we were most interested in the practical utility of the messages, and to keep the analyses consistent across outcomes, we opted for a between-participants approach in all analyses.

12. Reference 11 is the same as reference 6, and 65/67 are also duplicated. There may be others and this should be checked more closely.

Many thanks for spotting this. We now corrected all references.

Reviewer #2:

Remarks to the Author:

A 27-country test of communicating the scientific consensus on climate change

How to successfully convince the general public about the main cause and severity of climate change is one of the main societal challenges we're facing. The current manuscript provides a systematic, pre-registered approach to better understand this challenge.

The set up of the study is sound and explained clearly and in sufficient detail for replication purposes. The statistical analyses are appropriate for the design and research question and the results are interpreted clearly. A particularly strong point of the study is that the authors do not focus solely on Western countries: as climate change is a global challenge, it requires a global approach.

Thank you for the very detailed and thoughtful review of our work. We are very pleased that you find it to be of potential value and have attempted to provide both detailed responses to your comments as well as improvements to the manuscript that directly align with those. We hope you find these have produced an overall better study while also reassuring you about some concerns raised.

1. I do have one major concern, and this could be a 'deal breaker'. My major concern deals with how the data were collected: this has largely been done through convenience sampling. Even the selection of participating countries is some kind of convenience sample from the 'population' of all countries in the world. This raises serious concerns w.r.t. representativeness of the sample. Especially since one of the main goals is checking whether the proportion of the general population that agrees that climate change is

anthropogenic aligns with the scientific consensus percentage of 97%, representivity of the sample is a must.

By recruiting participants via snowballing, social media and mailing lists the risk of non-representativeness is large. The demographics in Table 2 also contain a number of indications that the sample is far from representative. The participants largely seem to be young (mean age 33.7) and highly educated (68% has university education). Furthermore in 14 out of 27 countries, the male/female ratio is so unbalanced that either men or women constitute at least 60% of the sample. If these demographics are biased, what's to say that the percentages as presented in e.g. Figure 1 aren't severely biased as well? That you adjust for gender and other demographics in the analysis is helpful but mitigates this problem only partially.

Even though the rest of the work seems outstanding and state of the art, the adagium 'garbage in, garbage out' hold true here. Without guarantees on the quality of the data, the quality of the results cannot be guaranteed...

We agree that considerations of sample representativeness are important. However, descriptively checking the proportion of people who correctly estimate the 97% consensus was not one of the primary goals of the study. We simply tested underestimation in each country's sample because theoretical accounts (e.g., the Gateway Belief Model, van der Linden, 2021) predict that underestimation and correction of misperceptions are the mechanisms behind this intervention. If most people in most samples already had correct perceptions of the scientific consensus, the intervention should theoretically not work. This was therefore more a test of precondition for the intervention to work rather than the main goal of the study. Due to convenience sampling, in this part of the manuscript, we talk about country samples rather than countries.

We also make the above point more explicit on p. 8: *These descriptives are unlikely to be representative of misperceptions per country due to our convenience sampling approach; rather they demonstrate that misperceptions are present in our samples, a prerequisite for consensus messaging to be effective.*

van der Linden, S. (2021). The Gateway Belief Model (GBM): A review and research agenda for communicating the scientific consensus on climate change. *Current Opinion in Psychology*, 42, 7–12. <https://doi.org/10.1016/j.copsyc.2021.01.005>

With regards to the experimental part of the study, several previous studies have shown that average treatment effects can be accurately estimated in experiments using convenience samples (Coppock, 2018; Mullinix et al., 2015). For example, Mullinix et al. (2015) showed across 20 experiments that nationally representative population-based samples and convenience samples (using MTurk) produce similar inferences regarding statistical significance and the direction of the treatment effect. Importantly, only

one out of 20 comparisons revealed a significantly different effect size estimate between the population-based and convenience sample. Moreover, controlling for socio-demographic variables in the analysis—as done in our study—can further reduce any potential differences in effect size estimates between population-based and convenience samples (Weinberg et al., 2014). Overall, substantial evidence shows that experimental treatment effects from convenience samples seem to produce similar results to nationally representative population-based samples. This is also supported by the fact that the effect size estimates in our study (misperception correction: $d = 0.47$; climate change attitudes: $d = 0.04-0.09$) closely align with effect size estimates from previous meta-analyses on scientific consensus messaging (misperception correction: $g = 0.56$; climate change attitudes: $g = 0.09-0.12$) that rely on mostly US-based studies with nationally representative samples.

We would also like to emphasize that hard-to-reach populations will most likely not be exposed to, and thus influenced by, a scientific consensus message if used in practice. Specifically, for scientific consensus messaging effects to be practically relevant, one needs to reach a large target audience, due to the relatively small effect sizes. This is likely to be achieved with, for example, online (social media) campaigns. These online campaigns are more likely to reach people who are more similar to our sample than the general population in each tested country, namely younger, more educated, and more liberal individuals that pay more attention to politics and that are more likely to be female (Marengo et al., 2020; Mellon & Prosser, 2017).

We address these points in the discussion (p. 15): *Third, social media users are generally younger, more educated, more liberal, more likely to be female, and pay more attention to politics^{69,70}, which is also reflected in our current samples. On the one hand, our social-media-based sampling approach may have led us to underestimate the intervention effects because, for example, younger and more educated individuals are more likely to believe in climate change⁷¹, which is, in turn, associated with higher perceptions of the scientific consensus⁷². On the other hand, several previous studies have shown that average treatment effects can be accurately estimated in experiments using convenience samples^{73,74}. This is also supported by the fact that the effect size estimates in our study (misperception correction: $d = 0.47$; climate change attitudes: $d = 0.04-0.09$) closely align with effect size estimates from previous meta-analyses on scientific consensus messaging (misperception correction: $g = 0.56$; climate change attitudes: $g = 0.09-0.12$) that rely on mostly US-based studies with nationally representative samples. From a practical perspective, hard-to-reach populations (e.g., people who do not have access to the internet or do not use social media) will most likely not be exposed to, and thus influenced by, a scientific consensus message when used by policymakers in, for example, online campaigns. We do not discount the importance of those populations; we simply highlight this consideration in the context of the effectiveness of this specific intervention.*

Coppock, A. (2019). Generalizing from survey experiments conducted on Mechanical Turk: A replication approach. *Political Science Research and Methods*, 7(3), 613-628.

Marengo, D., Sindermann, C., Elhai, J. D., & Montag, C. (2020). One social media company to rule them all: associations between use of Facebook-owned social media platforms, sociodemographic characteristics, and the big five personality traits. *Frontiers in Psychology*, 11, 527189.

Mellon, J., & Prosser, C. (2017). Twitter and Facebook are not representative of the general population: Political attitudes and demographics of British social media users. *Research & Politics*, 4(3), 2053168017720008.

Mullinix, K. J., Leeper, T. J., Druckman, J. N., & Freese, J. (2015). The generalizability of survey experiments. *Journal of Experimental Political Science*, 2(2), 109-138.

van der Linden, S. (2021). The Gateway Belief Model (GBM): A review and research agenda for communicating the scientific consensus on climate change. *Current Opinion in Psychology*, 42, 7–12. <https://doi.org/10.1016/j.copsyc.2021.01.005>

Weinberg, J. D., Freese, J., & McElhattan, D. (2014). Comparing data characteristics and results of an online factorial survey between a population-based and a crowdsourcing-recruited sample. *Sociological Science*, 1.

As said, the remainder of the manuscript is of excellent quality. I have a number of minor issues and questions for clarification:

3. Fig. 1: the countries are now sorted alphabetically. It is more insightful to sort them on basis of their score (thus either the red or blue percentage).

Thank you for raising this point. We thought about this as well when designing the figure. However, we decided against it because we do not want to overemphasize these cross-country differences as our samples are not representative. Sorting according to the extent of the misperceptions might invite these comparisons and invite misleading conclusions.

4. The differences between countries could be explored more. Fig.1 gives results per country, but this is not interpreted in detail for (systematic) differences. Do, for instance, the ~10 European countries yield systematically lower/higher results than e.g. the Asian or African countries? From all countries with a 'sufficient' sample size, China clearly scores lower than the rest; how could this be explained? Etc.

Thank you for this point. As it is closely related to the previous points about sample representativeness and descriptives, we believe it is not beneficial to emphasize descriptive cross-country differences on non-representative samples. However, following the suggestion of R1, we conducted additional analyses to detect whether differences in national culture (individualism-collectivism and power distance) could explain some heterogeneity in the effectiveness of the interventions (p. 13): *Following a suggestion from one of the reviewers, we also explore whether country-level characteristics, such as individualism-collectivism and power distance, moderate the effects of both interventions on perceptions of the reality consensus and crisis agreement. We find no convincing evidence for any moderation by these two cultural dimensions (Supplementary Results 2.6).*

We also paste below the results from the Supplement: *For the classic consensus message, we find moderate and strong evidence against any moderation of the effect on consensus perceptions by individualism-collectivism ($BF_{10} = 0.20$; $b = -0.05$, 95% CI [-0.11, 0.01]) and power distance ($BF_{10} = 0.08$; $b = 0.03$, 95% CI [-0.03, 0.10]) respectively. This was also the case for the updated consensus message: we find moderate evidence against a moderation by individualism-collectivism ($BF_{10} = 0.18$, $b = -0.04$, 95% CI [-0.10, 0.02]) and power distance ($BF_{10} = 0.16$, $b = 0.05$, 95% CI [-0.01, 0.11]) on consensus perceptions. For agreement perceptions changes in the updated (vs. control) condition, the results are inconclusive, with weak evidence against a moderation by individualism-collectivism ($BF_{10} = 0.35$, $b = -0.07$, 95% CI [-0.15, 0.02]) and weak evidence for a moderation by power distance ($BF_{10} = 1.53$, $b = 0.08$, 95% CI [0.02, 0.15]).*

Reviewer #3:

Remarks to the Author:

The authors are to be congratulated on a superbly written paper that is likely to be of wide interest within the climate change communication community. The literature review of the paper could serve as a model for graduate students to follow when they begin publishing. And the discussion was thoughtful and will be useful for strategists planning climate information campaigns.

Thank you very much for your thoughtful and detailed comments. We have attempted to provide thorough responses here along with the revised manuscript, and hope you find it to be a valuable improvement of the work.

1. My single concern about the study is its use of convenience samples, and the ways in which this may have affected the results. There were no quotas set to complete the survey, which would have improved the representativeness of the samples. Comparisons between Table 2 in the paper, which shows the demographics of each country's sample, and Table 3.1 in the supplementary materials, which shows population statistics for each country, reveal large discrepancies between the samples' and the nations' demographics. For example, in 25 of the 27 countries, more than half of the sample has a university degree, while the national data show that only 5 of the 27 countries have populations in which half have completed post-secondary education. In the Egyptian sample, 96% have a university degree, while nationally, only 13% have post-secondary degrees. Discrepancies are apparent in gender and urbanicity as well: Three-quarters of the samples in Georgia, Lebanon & Serbia are female; 92% of the Egyptian sample lives in urban areas, compared to 43% of the population.

Quotas would also have been helpful for political ideology: As right-leaning people have been shown to be more responsive to consensus messages, samples which do not reflect the national balance will not assess the effects of the messages accurately.

The demographic differences between the samples and nations may have affected the results in multiple ways. For example, if people with more education have more familiarity with the scientific consensus, the mean pre-intervention consensus perceptions shown in Figure 1 are overestimated for the nations with large sample-population discrepancies. And the reported effects of the messages are being underestimated, as the people who are unfamiliar with the messages are underrepresented. I believe discussion of this limitation of the research should be added to the paper.

Many thanks for raising this important point about sample representativeness and demographics. We now explicitly note our sampling approach in the abstract to increase transparency (*online convenience samples from 27 countries (N = 10,527)*), and thoroughly consider this as a limitation in the discussion (p. 15-16):

Third, social media users are generally younger, more educated, more liberal, more likely to be female, and pay more attention to politics^{69,70}, which is also reflected in our current samples. On the one hand, our social-media-based sampling approach may have led us to underestimate the intervention effects because, for example, younger and more educated individuals are more likely to believe in climate change⁷¹, which is, in turn, associated with higher perceptions of the scientific consensus⁷². On the other hand, several previous studies have shown that average treatment effects can be accurately estimated in experiments using convenience samples^{73,74}. This is also supported by the fact that the effect size estimates in our study (misperception correction: $d = 0.47$; climate change attitudes: $d = 0.04-0.09$) closely align with effect size estimates from previous meta-analyses on scientific consensus messaging (misperception correction: $g = 0.56$; climate change attitudes: $g = 0.09-0.12$) that rely on mostly US-based studies with

nationally representative samples. From a practical perspective, hard-to-reach populations (e.g., people who do not have access to the internet or do not use social media) will most likely not be exposed to, and thus influenced by, a scientific consensus message when used by policymakers in, for example, online campaigns. We do not discount the importance of those populations; we simply highlight this consideration in the context of the effectiveness of this specific intervention.

Despite this weakness, this is an excellent study overall that definitely merits publication.

Smaller points:

2. I believe the data from Lebanon should be deleted from the paper, as the sample is far too small to yield meaningful results ($n=9$); Tunisia's data is also questionable due to the small sample ($n=92$).

We did not preregister any exclusion criteria with regards to minimum sample size per country because the analytical approach precludes the chance that a small number of cases biases the conclusions in the overall sample. We agree that the data at the country level for Lebanon is not informative; however, country-specific conclusions are not the focus of our analyses and the paper in general. We therefore do not think it is necessary to delete the data from Lebanon or Tunisia, especially because they come from understudied populations on the topic.

To illustrate, we re-ran our H1a analysis, comparing consensus perceptions in the classic vs. control condition. No qualitative changes were detected when excluding Lebanon ($BF_{10} = 1.01 \times 10^{12}$, Cohen's $d = 0.4479$), compared to the full sample analysis ($BF_{10} = 2.01 \times 10^{12}$, Cohen's $d = 0.4491$).

3. I couldn't locate the OSF file cited in line 178; please add the file name to help readers find the data.

We added an explanation in that sentence: *For future meta-analyses, we provide a summary table of the results for each outcome per country on OSF (<https://bit.ly/44hKe7F>; subfolder outcome-summaries in the Data & Analyses tab).*

To clarify, there is a separate .csv file for each outcome to simplify readability, as we provide detailed descriptive statistics in addition to Cohen's d s in these files.

4. On lines 434-435, the authors state that the consensus message does not extend to support for public action. This should be amended to say the message does not extend DIRECTLY to policy support. The Gateway Belief Model does not posit a direct effect, which – as the authors note – their data cannot test.

Thank you for spotting this, we now added 'directly' to be more precise.

While overall, the writing is excellent, there were several sentences I found confusing:

5. Footnote 1

Rephrased: *We distinguish between reality consensus and crisis agreement to emphasize that scientific consensus on the reality of climate change was obtained by analyzing abstracts of scientific publications, while scientific agreement with regards to climate change as a crisis was obtained by surveying IPCC authors (i.e., percentage of IPCC authors who agree with the statement that climate change is a crisis).*

6. The sentence in lines 133-135

7. The sentence in lines 389-391

Rephrased: *While this decay in effectiveness is probable when people are not exposed to contrarian views, the information ecosystem contains climate misinformation^{64,65}, particularly in contexts where climate change is a politicized topic.*

8. The sentence in lines 467-469

Rephrased: *On social media, we posted in special interest groups that relate to current events, popular culture, or media discussions. We also posted comments on discussion threads of major news stories unrelated to climate change or sustainability.*

Decision Letter, first revision:

23rd February 2024

Dear Dr. Geiger,

Thank you for your patience as we've prepared the guidelines for final submission of your Nature Human Behaviour manuscript, "A 27-country test of communicating the scientific consensus on climate change" (NATHUMBEHAV-23072458A). Please carefully follow the step-by-step instructions provided in the attached file, and add a response in each row of the table to indicate the changes that you have made. Please also address the additional marked-up edits we have proposed within the reporting summary. Ensuring that each point is addressed will help to ensure that your revised manuscript can be swiftly handed over to our production team.

We would hope to receive your revised paper, with all of the requested files and forms within two-three weeks. Please get in contact with us if you anticipate delays.

If you have not done so already, please alert us to any related manuscripts from your group that are

under consideration or in press at other journals, or are being written up for submission to other journals (see: <https://www.nature.com/nature-research/editorial-policies/plagiarism#policy-on-duplicate-publication> for details).

Nature Human Behaviour offers a Transparent Peer Review option for new original research manuscripts submitted after December 1st, 2019. As part of this initiative, we encourage our authors to support increased transparency into the peer review process by agreeing to have the reviewer comments, author rebuttal letters, and editorial decision letters published as a Supplementary item. When you submit your final files please clearly state in your cover letter whether or not you would like to participate in this initiative. Please note that failure to state your preference will result in delays in accepting your manuscript for publication.

In recognition of the time and expertise our reviewers provide to Nature Human Behaviour's editorial process, we would like to formally acknowledge their contribution to the external peer review of your manuscript entitled "A 27-country test of communicating the scientific consensus on climate change". For those reviewers who give their assent, we will be publishing their names alongside the published article.

Cover suggestions

We welcome submissions of artwork for consideration for our cover. For more information, please see our guide for cover artwork.

ORCID

Non-corresponding authors do not have to link their ORCIDs but are encouraged to do so. Please note that it will not be possible to add/modify ORCIDs at proof. Thus, please let your co-authors know that if they wish to have their ORCID added to the paper they must follow the procedure described in the following link prior to acceptance: <https://www.springernature.com/gp/researchers/orcid/orcid-for-nature-research>

Nature Human Behaviour has now transitioned to a unified Rights Collection system which will allow our Author Services team to quickly and easily collect the rights and permissions required to publish your work. Approximately 10 days after your paper is formally accepted, you will receive an email in providing you with a link to complete the grant of rights. If your paper is eligible for Open Access, our Author Services team will also be in touch regarding any additional information that may be required to arrange payment for your article.

Please note that *Nature Human Behaviour* is a Transformative Journal (TJ). Authors may publish their research with us through the traditional subscription access route or make their paper immediately open access through payment of an article-processing charge (APC). Authors will not be required to

make a final decision about access to their article until it has been accepted. Find out more about Transformative Journals

Authors may need to take specific actions to achieve compliance with funder and institutional open access mandates. If your research is supported by a funder that requires immediate open access (e.g. according to Plan S principles) then you should select the gold OA route, and we will direct you to the compliant route where possible. For authors selecting the subscription publication route, the journal's standard licensing terms will need to be accepted, including self-archiving policies. Those licensing terms will supersede any other terms that the author or any third party may assert apply to any version of the manuscript.

[REDACTED]

Best regards,
[REDACTED]

On behalf of

[REDACTED]

Reviewer #1:

Remarks to the Author:

As a reviewer of the original submission, I first read this revision "fresh" before the rebuttal. In this fresh reading I found it easier to follow and understand, and the main points were generally clearer. I also appreciated the additional descriptive information provided (e.g., the proportion of samples with estimates below the consensus; dropout rates; effect sizes by country) and additional analyses (e.g., country-level explanations). So as an overall evaluation I found this revision much improved.

In this new reading there were two issues that I thought were not as convincingly addressed, and both were related to my initial review. I am raising them again because they remain a concern for me, but I also accept that they may not be critical to the ultimate publication of the manuscript.

1. I continue to disagree with the authors about the importance of the main outcome measure – reality consensus, because it is very close conceptually to the manipulation. For me this is reinforced by the finding that the classic message was much more effective for those who were less familiar with this information (H4a) (p.12). Expressed differently, for people less likely to know the consensus, they will be more likely to report 97% because that is the main information they have and what they've just been told. In the author(s) response they note that 45% selected 97% in the experimental conditions, but the follow-up question is how many of these could be considered as low familiarity. But

perhaps this is getting too detailed.

The broader point is that as the manuscript is focused on practical rather than theoretical advances, to me the much more important outcomes for addressing climate change (the ultimate aim of this research) are the more distal ones, and most importantly support for action. Here, and for antecedents of action like worry, the effect sizes are null or very small – to me that LACK of association between climate change messaging (classic/updated) and support for public action is the main takeaway message of this paper. I note that the authors called some of the very weak effects as “rather notable” because the messaging does not directly target people’s beliefs or actions, but to me that means that these are some of the least important beliefs about climate change to target for influence, despite the Gateway Belief Model claims.

However, I accept that the authors have responded to these points, that these findings are likely to still be interest to some, and it’s the author(s) prerogative to focus on what they think is most interesting.

2. I appreciate that some cross-cultural coverage/analysis has now been included, but the level of engagement with culture still seems a bit cursory for one of the main contributions of the manuscript (data from 27 countries). This lack of attention continues to come through in the introduction, such as the statement: : “...if consensus messaging is ineffective when tested more globally—especially among skeptical or opposed groups—it would signal the limits of messaging interventions and the need to focus on different strategies to mobilize support for climate action” (p. 5). This portrays portray universal (global) effects as the only outcome of value - only a framing approach that’s effective globally is worth attention and pursuing.

To me this focus undersells the potential cultural component – the research has the potential to understand whether some frames could work better in some countries than others, allowing for country-level tailoring of messages (and potentially generalize to those with similar characteristics if cultural level factors like Individualism and Power Distance are predictors). This is a strength of the research even if the results do not find such cultural differences. It would be good if the author(s) could make a clearer case for the potential consequences of identifying cultural differences in the introduction.

I appreciate the author(s) inclusion of country-level analyses for individualism and power distance, but as exploratory analyses it seems like they did the minimum requested when there is a cultural conundrum that could be explored further. As the author(s) note, for consensus messaging “these analyses also show extremely strong support for between-country heterogeneity” (p. 10), and while the two dimensions analyzed did not account for differences, that seemed a very early point to give up on trying to explain country differences. For instance, there are many other possible explanations ranging from other cultural dimensions to environmental performance (e.g., the Yale Environmental Performance Index). It seemed like a bit of a missed opportunity to use these hard-to-get country samples without doing a more comprehensive exploration of country differences where these were identified.

But as for the previous point, even though I think explaining country heterogeneity is important, I appreciate that this is not the author(s) focus and they have to make decisions about what to cover in the manuscript.

Minor point:

It was not clear how the paragraph at the top of p. 3 (Although bridging...) advanced the argument – it reads as if the authors are arguing that the research is unimportant. I only knew this was a link to crisis framing because I had read the previous manuscript. It would help to make the link to crisis in this paragraph, preferably at the beginning, otherwise readers will be left unclear about the main point of this paragraph.

Reviewer #2:

Remarks to the Author:

Thanks for the revision. I have no further comments

Reviewer #3:

Remarks to the Author:

My thanks to the authors for their careful and comprehensive responses to the reviewers' comments. I'm satisfied with the authors' responses and recommend that the article be published.

My one remaining request is a reorganization of the section in the results section on misperceptions of the consensus: The new sentence the authors have added to the final paragraph of this section stating that the results are not representative should be moved to the first paragraph. Readers should be forewarned as they read this section that the consensus estimates are only useful as prerequisites for the experiment.

Author Rebuttal, first revision:

REVIEWER COMMENTS:

Reviewer #1

As a reviewer of the original submission, I first read this revision “fresh” before the rebuttal. In this fresh reading I found it easier to follow and understand, and the main points were generally clearer. I also appreciated the additional descriptive information provided (e.g., the proportion of samples with estimates below the consensus; dropout rates; effect sizes by country) and additional analyses (e.g., country-level explanations). So as an overall evaluation I found this revision much improved.

Many thanks for your time in reviewing our submission again and your positive overall assessment of the manuscript.

In this new reading there were two issues that I thought were not as convincingly addressed, and both were related to my initial review. I am raising them again because they remain a concern for me, but I also accept that they may not be critical to the ultimate publication of the manuscript.

1. I continue to disagree with the authors about the importance of the main outcome measure – reality consensus, because it is very close conceptually to the manipulation. For me this is reinforced by the finding that the classic message was much more effective for those who were less familiar with this information (H4a) (p. 12). Expressed differently, for people less likely to know the consensus, they will be more likely to report 97% because that is the main information they have and what they've just been told. In the author(s) response they note that 45% selected 97% in the experimental conditions, but the follow-up question is how many of these could be considered as low familiarity. But perhaps this is getting too detailed.

The broader point is that as the manuscript is focused on practical rather than theoretical advances, to me the much more important outcomes for addressing climate change (the ultimate aim of this research) are the more distal ones, and most importantly support for action. Here, and for antecedents of action like worry, the effect sizes are null or very small – to me that LACK of association between climate change messaging (classic/updated) and support for public action is the main takeaway message of this paper. I note that the authors called some of the very weak effects as “rather notable” because the messaging does not directly target people’s beliefs or actions, but to me that means that these are some of the least important beliefs about climate change to target for influence, despite the Gateway Belief Model claims.

However, I accept that the authors have responded to these points, that these findings are likely to still be interest to some, and it’s the author(s) prerogative to focus on what they think is most interesting.

Thank you for raising these points. As for the first point about the intersection of familiarity and selecting exact percentage points, while this might be interesting to explore further, this article concerns group-level differences between the messaging conditions.

The broader point prompted us to go over the manuscript with specific focus on framing the (relative) importance of the outcomes. We now omit the ‘notable’ characterization of the identified effects from the discussion (p. 12): *“Second, this study finds nominally small effects⁶⁶ of scientific consensus messaging. However, these effects are in line with previous research^{26,27} and can be practically relevant⁶⁷, as the intervention is easily scalable to reach many people due to its brevity. Targeting specific subgroups, such*

as those on the political right who are most likely to underestimate the consensus, might also increase its effectiveness.”

More generally, we failed to identify referring to consensus perceptions as the most important outcome(s). What is more, the updated consensus message condition was conceptualized with an explicit goal and expectation that it would be beneficial for increasing climate change worry and support for public action (p. 3): “As such, an updated message that emphasizes the negative impacts of climate change and implies the need for public action might prove more effective at increasing belief in climate change as a crisis, climate change worry, and support for public action than the classic message²⁸. This might be especially useful in contexts where the public consensus on the reality of climate change is high, but a significant proportion still doubts the urgency of climate action²⁸.”

As for moderator analyses, we conducted these on consensus perceptions for two reasons. First, testing boundary conditions for consensus perceptions is theoretically important as it directly tests belief updating processes. Second, from a practical standpoint of statistical power, these tests were the most likely to yield reliable conclusions on boundary conditions due to the relatively large main effects.

We generally agree there are other important (personal) beliefs to target to boost climate change action. However, as we also argued in our previous response, personal beliefs about fundamental scientific facts are not trivial outcomes – given that they represent an alignment of people's attitudes with the scientific reality and thus enable people to base their consequent decisions based on a better-informed attitude. Indeed, such basic climate change beliefs are important determinants of pro-environmental behaviours (e.g., Kácha et al., 2022; McCrea et al., 2015).

References:

Kácha, O., Vintr, J., & Brick, C. (2022). Four Europes: Climate change beliefs and attitudes predict behavior and policy preferences using a latent class analysis on 23 countries. *Journal of Environmental Psychology*, 81, 101815. <https://doi.org/10.1016/j.jenvp.2022.101815>

McCrea, R., Leviston, Z., Walker, I., & Shyy, T.-K. (2015). Climate Change Beliefs Count: Relationships With Voting Outcomes at the 2010 Australian Federal Election. *Journal of Social and Political Psychology*, 3(1), 124–141. <https://doi.org/10.5964/jspp.v3i1.376>

2. I appreciate that some cross-cultural coverage/analysis has now been included, but the level of engagement with culture still seems a bit cursory for one of the main contributions of the manuscript (data from 27 countries). This lack of attention continues to come through in the introduction, such as the statement: : “...if consensus messaging is ineffective when tested more globally—especially among skeptical or opposed groups—it would signal the limits of messaging interventions and the need to focus on different strategies to mobilize support for climate action” (p. 5). This portrays universal (global) effects as the only outcome of value - only a framing approach that’s effective globally is worth attention and pursuing.

To me this focus undersells the potential cultural component – the research has the potential to understand whether some frames could work better in some countries than others, allowing for country-level tailoring of messages (and potentially generalize to those with similar characteristics if cultural level factors like Individualism and Power Distance are predictors). This is a strength of the research even if the results do not find such cultural differences. It would be good if the author(s) could make a clearer case for the potential consequences of identifying cultural differences in the introduction.

I appreciate the author(s) inclusion of country-level analyses for individualism and power distance, but as exploratory analyses it seems like they did the minimum requested when there is a cultural conundrum that could be explored further. As the author(s) note, for consensus messaging “these analyses also show extremely strong support for between-country heterogeneity” (p. 10), and while the two dimensions analyzed did not account for differences, that seemed a very early point to give up on trying to explain country differences. For instance, there are many other possible explanations ranging from other cultural dimensions to environmental performance (e.g., the Yale Environmental Performance Index). It seemed like a bit of a missed opportunity to use these hard-to-get country samples without doing a more comprehensive exploration of country differences where these were identified.

But as for the previous point, even though I think explaining country heterogeneity is important, I appreciate that this is not the author(s) focus and they have to make decisions about what to cover in the manuscript.

Thank you for raising this thoughtful point. We have now revised the cited part of the introduction to better reflect the importance of a multi-country dataset, with the final sentence as a direct response to the raised concern (p. 4): “On a translational level, a messaging approach that is effective across diverse contexts and audiences would provide a general guideline for climate change communication and could thus facilitate a more rapid move toward urgently-needed climate policies. If the effectiveness varies according to individual and country-level characteristics, this could inform targeting specific audiences within countries and/or calibrating consensus messaging interventions to different country contexts. However, if consensus messaging is ineffective when tested across a diverse set of countries, it would signal the limits of this intervention and the need to focus on different strategies to mobilize support for climate action.”

Furthermore, we now explicitly state that we do not have enough power to detect cross-level interactions, as evidenced by only weak evidence for or against a moderation effect in our exploratory analyses reporting (p. 10): *“However, we detect only weak evidence against or for any country-level moderation effects, which suggests that this study is underpowered to robustly probe such moderations. These results should, therefore, be seen as tentative and followed up by analyses on datasets including more countries”*, as well as in the discussion, where we also explicitly address the value of testing country-level predictors more broadly (p. 12): *“Fourth, we are unable to draw definitive conclusions about the extent of between-country heterogeneity and make concrete per-country recommendations about where scientific consensus messaging might be most effective, due to the convenience sampling approach and insufficient statistical power to detect moderations by country-level predictors (i.e., cultural dimensions). We encourage future research to continue testing message effectiveness within countries using representative samples and, possibly, our materials and translations, to ultimately make practical recommendations for climate change communication tailored to specific countries (for example, as previously done in Germany⁷³). In addition, datasets including a larger number of countries are essential for robustly testing country-level factors that might determine consensus messaging effectiveness.*

Minor point:

It was not clear how the paragraph at the top of p. 3 (Although bridging...) advanced the argument – it reads as if the authors are arguing that the research is unimportant. I only knew this was a link to crisis framing because I had read the previous manuscript. It would help to make the link to crisis in this paragraph, preferably at the beginning, otherwise readers will be left unclear about the main point of this paragraph.

Thank you for spotting this, we agree this was disrupting the flow of the introduction. We have now revised this transition (p. 2-3): *“Beyond the consensus on the reality of human-caused climate change^{5,24,25,28,33,35}, climate experts emphasize very high certainty of the adverse consequences of climate change and the urgency of climate action to curb these impacts in the 6th IPCC report³⁵. In line with this, 88% of surveyed IPCC authors reported that they think climate change constitutes a crisis³⁶. To align communication about the scientific consensus with these more up-to-date climate science assessments and potentially improve its effectiveness, we test supplementing the*

97% consensus that human-caused climate change is happening with the 88% agreement that climate change is an urgent matter (i.e., a crisis).”

Reviewer #3

My thanks to the authors for their careful and comprehensive responses to the reviewers' comments. I'm satisfied with the authors' responses and recommend that the article be published.

Many thanks for your time in reviewing our submission again and your positive assessment of the manuscript.

My one remaining request is a reorganization of the section in the results section on misperceptions of the consensus: The new sentence the authors have added to the final paragraph of this section stating that the results are not representative should be moved to the first paragraph. Readers should be forewarned as they read this section that the consensus estimates are only useful as prerequisites for the experiment.

Thank you, we have now moved this information to the beginning of the section (p. 6): *“In this section, we provide misperceptions of the reality consensus and crisis agreement per country prior to message exposure. These descriptives are unlikely to be representative of misperceptions per country due to the convenience sampling approach. Instead, they demonstrate that misperceptions are present in our samples—a prerequisite for consensus messaging to be effective.”*

Final Decision Letter:

Dear Sandra,

We are pleased to inform you that your Article "A 27-country test of communicating the scientific consensus on climate change", has now been accepted for publication in *Nature Human Behaviour*.

Please note that *Nature Human Behaviour* is a Transformative Journal (TJ). Authors may publish their research with us through the traditional subscription access route or make their paper immediately open access through payment of an article-processing charge (APC). Authors will not be required to make a final decision about access to their article until it has been accepted. Find out more about Transformative Journals

Authors may need to take specific actions to achieve compliance with funder and institutional open access mandates. If your research is supported by a funder that requires immediate open access (e.g.

according to Plan S principles) then you should select the gold OA route, and we will direct you to the compliant route where possible. For authors selecting the subscription publication route, the journal's standard licensing terms will need to be accepted, including self-archiving policies. Those licensing terms will supersede any other terms that the author or any third party may assert apply to any version of the manuscript.

Once your manuscript is typeset and you have completed the appropriate grant of rights, you will receive a link to your electronic proof via email with a request to make any corrections within 48 hours. If, when you receive your proof, you cannot meet this deadline, please inform us at risproduction@springernature.com immediately. Once your paper has been scheduled for online publication, the Nature press office will be in touch to confirm the details.

To assist our authors in disseminating their research to the broader community, our SharedIt initiative

provides you with a unique shareable link that will allow anyone (with or without a subscription) to read the published article. Recipients of the link with a subscription will also be able to download and print the PDF.

With best regards,
[REDACTED]